# HIGHER-ORDER STRUCTURE PREDICTION IN EVOLVING GRAPH SIMPLICIAL COMPLEXES

## ABSTRACT

Dynamic graphs are rife with higher-order interactions, such as co-authorship relationships and protein-protein interactions in biological networks, that naturally arise between more than two nodes at once. In spite of the ubiquitous presence of such higher-order interactions, limited attention has been paid to the higher-order counterpart of the popular pairwise link prediction problem. Existing higher-order structure prediction methods are mostly based on heuristic feature extraction procedures, which work well in practice but lack theoretical guarantees. Such heuristics are primarily focused on predicting links in a static snapshot of the graph. Moreover, these heuristic-based methods fail to effectively utilize and benefit from the knowledge of latent substructures already present within the higher-order structures. In this paper, we overcome these obstacles by capturing higher-order interactions succinctly as *simplices*, model their neighborhood by face-vectors, and develop a nonparametric kernel estimator for simplices that views the evolving graph from the perspective of a time process (i.e., a sequence of graph snapshots). Our method substantially outperforms several baseline higher-order prediction methods. As a theoretical achievement, we prove the consistency and asymptotic normality in terms of Wasserstein distance of our estimator using Stein's method.

## 1    INTRODUCTION

Numerous types of networks like social (Liben-Nowell & Kleinberg, 2007a), biological (Airoldi et al., 2006), and chemical reaction networks (Wegscheider, 1911) are highly dynamic, as they evolve and grow rapidly via the appearance of new interactions, represented as the introduction of new links / edges between the nodes of a network. Identifying the underlying mechanisms by which such networks evolve over time is a fundamental question that is not yet fully understood. Typically, insight into the temporal evolution of networks has been obtained via a classical inferential problem called *link prediction*, where given a snapshot of the network at time $t$ along with its linkage pattern, the task is to assess whether a pair of nodes will be linked at a later time $t' > t$.

While inferring pairwise links is an important problem, it is oftentimes observed that most of the real-world graphs exhibit *higher-order group-wise interactions* that involve more than two nodes at once. Examples illustrating human group behavior involve a co-author relationship on a single paper and a network of e-mails to multiple recipients. In nature too, one can observe several proteins interacting together in a biological network simultaneously.

In spite of their significance, in comparison to single edge inference, relatively fewer works have studied the problem of predicting higher-order group-wise interactions. Benson et al. (2018) originally introduced a *simplex* to model group-wise interactions between nodes in a graph. They proposed predicting a *simplicial closure event*, whereby an *open simplex* (with just pairwise interactions between member vertices) transitions to a *closed simplex* (where all member vertices participate in the higher-order relationship simultaneously), in the near future. Figure 1 (Middle) shows an example of such a transition from an *open triangle* to a *closed* one. Recently, several works have proposed modeling higher-order interactions as *hyperedges* in a hypergraph (Xu et al., 2013; Zhang et al., 2018; Yoon et al., 2020; Patil et al., 2020). Given a hyperedge $h_t$ at time $t$, the inference task is to predict the future arrival of a new hyperedge $h_{t'}$, which covers a larger set of vertices than $h_t$ and contains all the vertices in $h_t$. Figure 1 (Right) illustrates this hyperedge prediction task.

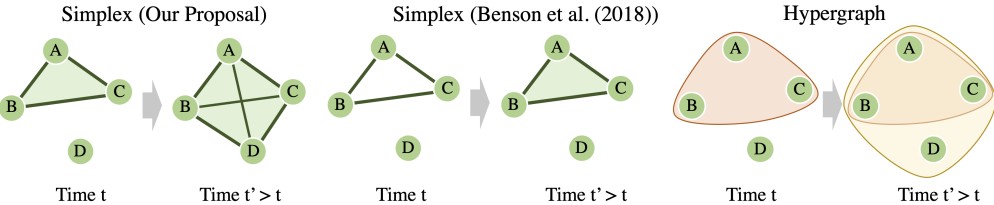

Figure 1: **[Left]** Given a 4-node graph, at time $t$, the 2-simplex $[A, B, C]$ also contains 1-simplices $[A, B], [B, C]$ and $[A, C]$. At time $t' > t$, the 2-simplex *evolves* (by connecting with $D$) to a 3-simplex $[A, B, C, D]$ which additionally contains 1-simplices $[A, D], [B, D]$ and $[C, D]$, along with 2-simplices $[A, B, D], [A, C, D]$ and $[B, C, D]$. **[Middle]** Simplex setting with Benson et al. (2018). The method predicts $[A, B, C]$ (*closed triangle*) at time $t' > t$ from an *open triangle* with links/edges $[A, B], [B, C]$ and $[A, C]$ at time $t$. **[Right]** Hypergraph represents a triple $[A, B, C]$ as a hyperedge, without any of its subsets. It cannot distinguish between $[[A, B], [B, C], [A, C]]$ and $[A, B, C]$.

Although prediction models based on either *simplicial closure event* prediction or *hyperedge* arrival, deal with higher-order structures, they both fail to capture the *highly complex* and *non-linear evolution* of higher-order structures over time. Both these kinds of models have two major limitations. First, they predict structures from a *single static snapshot* of the graph, thus not viewing the evolution process of adding new edges as a time process. Second, their feature extraction is mostly based on popular heuristics (Adamic & Adar, 2003; Brin & Page, 2012; Jeh & Widom, 2002; Zhou et al., 2009a; Barabási & Albert, 1999; Bhatia et al., 2019) that work well in practice but are not accompanied by strong theoretical guarantees. In addition to the aforementioned shortcomings, hypergraph based methods model higher-order structures as hyperedges, which omit *lower-dimensional substructures* present within a single hyperedge. As a consequence, they cannot distinguish between various substructure relationships. For example, hyperedge $[A, B, C]$ in Figure 1 (Right) cannot distinguish between group relationships like $[[A, B], [B, C], [A, C]]$ (a set of pairwise interactions) versus $[A, B, C]$ (all $A$, $B$ and $C$ simultaneously in a relationship). This problem is remedied by the use of simplices because they naturally model these substructures as a collection of subsets (i.e., faces) of the simplex.

We provide real-world examples of where our simplicial complex based approach can play a significant role. **(i) Organic Chemistry:** It is quite common to have the *same* set of elements interacting with each other in different configurations, which result in very different functioning compounds (Ma et al., 2011). Specifically, R-thalidomide and S-thalidomide are two different configurations of thalidomide, where the R-form was meant to help sedate pregnant women, while the S-form unfortunately resulted in birth defects. This is a famous example in stereo chemistry to show the consequences of mistaking two extremely close configurations (differing by a single bond) as being the same. Structure prediction to avoid such phenomenon in drug synthesis allows chemists to achieve a much higher yield and avoid wastage of expensive resources. **(ii) Gene expression networks**: Gene networks have nodes that represent genes and edges connect genes which have similar expression patterns (Zhang & Horvath, 2005). Subgraphs called *modules* are tightly connected genes in such a gene expression network. Genomics research provides evidence that higher-order gene expression relationships (like second and third-order) and their measurements can have very important implications for cancer prognosis. When making structural predictions in these aforementioned examples, our simplicial complex based approach provides much more fine-grained control over competing methods by capturing subtler differences in configurations.

To combat these challenges, our approach views the evolving graph[1] as a *time process* under the framework of nonparametric time series prediction, which models the evolution of higher-order structures (as simplices) and their *local neighborhoods* (spatial dimension) over a moving time window (temporal dimension). Our inference problem is then modeled as predicting the evolution to a higher-dimensional simplex at time $t' > t$, given a simplex at time $t$. It is important to note that this task is *more general* and *greatly diverges* from the task proposed by Benson et al. (2018). Our

---

[1]We handle the *incremental model* (edge insertions only) as opposed to the harder *fully dynamic model* (edge insertions and deletions allowed) for which most previous methods too cannot provide theoretical guarantees.

task requires just a single simplex $\sigma$ in order to predict a higher-dimensional simplex $\tau$, whose *face / subset* is $\sigma$, whereas Benson et al. (2018) requires the presence of *all* constituent $\sigma$ faces in order to predict $\tau$. For example, in Benson et al. (2018) (also shown in Figure 1 (Middle)) all faces $[A, B]$, $[B, C]$ and $[A, C]$ (open triangle) need to be present in order to predict a closed triangle $[A, B, C]$. Contrastingly, in our approach, just a single face/edge like $[A, B]$ or $[B, C]$ or $[A, C]$, suffices to predict its evolution to $[A, B, C]$. Figure 1 (Left) illustrates an additional example of our proposal to predict a 3-simplex $[A, B, C, D]$ given only one of its faces $[A, B, C]$.

To this effect, we succinctly capture the features characterizing the local neighborhood of a simplex as a combination of a *face-vector* (Björner & Kalai, 2006), which is a well-established vector signature in combinatorial topology literature and a novel *scoring function* which infers the affinity of sub-simplices based on the strength of their past interactions. Based on these features, we design a *kernel estimator* to infer future evolution to higher-dimensional simplices and prove both the consistency and asymptotic normality of our estimator.

**Our contributions:** (a) We propose a kernel estimator that predicts higher-order interactions in an evolving network. (b) We prove the consistency and asymptotic normality of our kernel estimator. (c) We evaluate our method on real-world dynamic networks by proposing higher-order link prediction baselines and observe significant gains in prediction accuracy in comparison to the baselines.

### 1.1 RELATED STUDIES

**Single link prediction:** Most literature that predicts a single edge/link can be broadly classified as based on: (i) heuristics, (ii) random-walks, or (iii) graph neural networks (GNNs). (i) Heuristic methods comprise of Common neighbors, Adamic-adar (Adamic & Adar, 2003), PageRank (Brin & Page, 2012), SimRank (Jeh & Widom, 2002), resource allocation (Zhou et al., 2009a), preferential attachment (Barabási & Albert, 1999), persistence homology based ranking (Bhatia et al., 2019), and similarity-based methods (Liben-Nowell & Kleinberg, 2007b; Lü & Zhou, 2011). (ii) Random walk based methods consist of DeepWalk (Perozzi et al., 2014) , Node2Vec (Grover & Leskovec, 2016b) and SpectralWalk (Sharma et al., 2020). (iii) Finally, for both link prediction and node classification tasks, recent works are mainly GNN-based methods such as VGAE (Kipf & Welling, 2016), WYS (Abu-El-Haija et al., 2018), and SEAL (Zhang & Chen, 2018b).

**Higher-order link prediction:** Benson et al. (2018) are the first to introduce a *higher-order link prediction problem* where they study the likelihoods of future higher-order group interactions as *simplicial closure* events (explained earlier). Furthermore, there are studies using *hypergraphs* which also help naturally represent group relations (Xu et al., 2013; Zhang et al., 2018; Yoon et al., 2020; Patil et al., 2020). Especially, to represent higher-order relationships, Yoon et al. (2020) proposed $n$-projected graphs. For larger $n$, i.e., higher-order groups, the enumeration of subsets, and keeping track of node co-occurrences quickly becomes infeasible. In comparison to a hypergraph, our *graph simplicial complex* is closed under taking subsets, which enables us to better encode more information for improved inference.

## 2 PRELIMINARY: GRAPH SIMPLICIAL COMPLEX (GSC)

We start with a general notion of an *abstract simplicial complex* (ASC), then define a *simplex* using ASCs. We specialize this definition to graphs and define a *graph simplicial complex* (GSC).

**Definition 1** (Abstract simplicial complex and simplex)**.** An *abstract simplicial complex* (ASC) is a collection $A$ of finite non-empty sets, such that if $\sigma$ is an element of $A$, then so is every non-empty subset of $\sigma$. The element $\sigma$ of $A$ is called a *simplex* of $A$; its *dimension* is one less than the number of its elements.

Now, we analyze graphs using the definition of ASCs. Let $G = (V, E)$ be a finite graph with

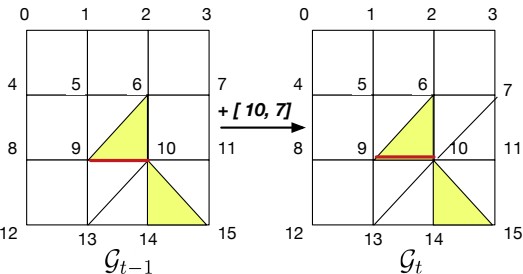

Figure 2: Example of evolution of GSC $\mathcal{G}$. The yellow triangles are 2-simplices. In the $k$-ball around 1-simplex $[9, 10]$ (in red), a 1-simplex $[10, 7]$ is added at time $t$.

vertex set $V$ and edge set $E$. A *graph simplicial complex (GSC)* $\mathcal{G}$ on $G$ is an ASC consisting of subsets of $V$. In particular, $\mathcal{G}$ is a *collection of subgraphs of G*. With graphs, we denote a $d$-dimensional simplex (or $d$-simplex) of a GSC by $\sigma^{(d)} = [v_0, v_1, \ldots, v_d]$. Each non-empty subset of $\sigma^{(d)}$ is called a *face* of $\sigma^{(d)}$.

We define several notions related to GSCs, that are useful for describing the evolution of graphs.

**Definition 2** (Filtered GSC). For $I \subset \mathbb{N}$, a *filtered GSC* indexed over $I$ is a family $(\mathcal{G}_t)_{t \in I}$ of GSCs such that for every $t \leq t'$ in $I$, $\mathcal{G}_t \subset \mathcal{G}_{t'}$ holds.

Obviously, $\mathcal{G}_{t_0} \subset \mathcal{G}_{t_1} \subset \ldots \mathcal{G}_{t_n}$ is a discrete filtration induced by the arrival times of the simplices: $\mathcal{G}_{t_i} \setminus \mathcal{G}_{t_{i-1}} = \sigma_{t_i}$. This depicts a higher-order analogue of an evolving graph (incremental model), which allows *attaching* new simplices at each time-step to an existing GSC to build a new GSC. A filtered GSC $\mathcal{G}_{t,p}$ for the last $p$ discrete time steps is defined as $\mathcal{G}_{t,p} := (\mathcal{G}_{t'})_{t'=t-p}^{t} = (\mathcal{G}_{t-p}, \ldots, \mathcal{G}_t)$, where $\mathcal{G}_{t'} \supset \mathcal{G}_{t'-1}$.

We define a notion for dealing with the neighborhood around a given simplex $\sigma^{(d)} \in \mathcal{G}$. We introduce a set of all simplices of dimension $d'$ or less from $\mathcal{G}$, i.e., $\mathcal{G}_-^{(d')} := \{\sigma^{(d)} \in \mathcal{G} \mid d \leq d'\}$. $\mathcal{G}_-^{(0)}$ is a vertex set and $\mathcal{G}_-^{(1)}$ is the set of edges and vertices. We also write $i \sim j$ whenever vertices $i$ and $j$ are adjacent in $\mathcal{G}_-^{(1)}$, and write $i \sim_k j$ to indicate that vertex $j$ is $k$-*reachable* from $i$, i.e., there exists a path of length at most $k$, connecting $i$ and $j$ in $\mathcal{G}_-^{(1)}$. Then, we define a ball around $\sigma^{(d)}$.

**Definition 3** ($k$-ball centered at vertex and simplex). At time $t$, we define a $k$-*ball centered at vertex* $i$ by $B_{t,k}(i) := \{j : i \sim_k j \text{ and } i, j \in \mathcal{G}_{t-}^{(0)}\}$. We also define a $k$-*ball centered at a simplex* $\sigma^{(d)}$ as $B_{t,k}(\sigma^{(d)}) := \bigcup_{i:\text{Vert}(\sigma^{(d)})} B_{t,k}(i)$, where $\text{Vert}(\sigma^{(d)})$ denotes all vertexes in $\sigma^{(d)}$.

Now, we define a *sub-complex* $\mathcal{G}_t'(\sigma^{(d)}) \subseteq \mathcal{G}_t$ as the GSC that contains all the simplices in $\mathcal{G}_t$ spanned by the vertices in the $k$-ball $B_{t,k}(\sigma^{(d)})$.

## 3 PREDICTING THE ARRIVAL OF HIGHER-ORDER SIMPLICES

We consider the prediction of a simplex's arrival in the setting described below. Consider a filtered GSC $\mathcal{G}_{t,p}$. At time $t$, given a $d$-dimensional simplex $\sigma^{(d)} = [v_0, \cdots, v_d]$, we predict the formation of a $(d+1)$-simplex $\tau^{(d+1)} = [v_0, \cdots, v_d, \widetilde{v}]$ with a new vertex $\widetilde{v} \in B_{t,k}(\sigma^{(d)})$. Here, we restrict $\widetilde{v}$ to be $k$-reachable from $\sigma^{(d)}$. In order to find out which simplices and vertices are most likely to appear in $\tau^{(d+1)}$ at time $t+1$, we need to design features for $\sigma^{(d)}$ and $\widetilde{v}$.

### 3.1 FEATURE DESIGN FOR SIMPLEX

We develop a feature design of a simplex associated with the notion of $k$-balls. The design is organized into two main elements: (i) a face-vector with a sub-complex, and (ii) a scoring function.

*(i) Face vector of sub-complex*: We first define a face-vector of a fixed GSC. The face-vector is an important *topological invariant*[2] of the GSC.

**Definition 4** (face-vector). A combinatorial statistic of $\mathcal{G}$ is the face-vector (or $f$-vector) as $f(\mathcal{G}) = (f_{-1}, f_0, \ldots, f_{d-1})$, where $f_k = f_k(\mathcal{G})$ records the number of $k$-dimensional faces $\sigma^{(k)} \in \mathcal{G}$.

We then define the feature of a simplex $\sigma^{(d)}$ at time $t$, denoted by $N_t(\sigma^{(d)}) = f(\mathcal{G}_t'(\sigma^{(d)}))$. In words, the feature is compactly represented as the face-vector of sub-complex $\mathcal{G}_t'(\sigma^{(d)})$. Our face-vector representation of a node's neighborhood can be considered as a *higher-order analogue* of the *Weisfeiler-Lehman* (WL) kernel Shervashidze et al. (2011) on unlabeled graphs, which for each vertex, iteratively aggregates the vertex degrees of its immediate neighbors to compute a unique vector of the target vertex that captures the structure of its extended neighborhood.

*(ii) Scoring function:* The purpose of this function is to extract the features of $\widetilde{v}$ using its proximity to $\sigma^{(d)}$. To this end, we begin by describing *affinity* between two vertices. Given two vertices

---

[2]A topological invariant is a property that is preserved by *homeomorphisms*.

Table 1: Notation table

**Basic**

| | |
|---|---|
| $G = (V, E)$ | graph with vertex set $V$ and edge set $E$ |
| $\mathcal{G}$ | graph simplicial complex (collection of subgraphs of $G$) |
| $\sigma^{(d)} = [v_0, ..., v_d]$ | $d$-dimensional simplex ($d$-simplex) |
| $\tau^{(d+1)} = [v_0, ..., v_d, \widetilde{v}]$ | $d + 1$-simplex for prediction |
| $\mathcal{G}_{t,p} = \{\mathcal{G}_{t-p}, ..., \mathcal{G}_t\}$ | GSCs from the previous $p$ time steps |
| $\mathcal{G}_{-}^{(d')} = \{\sigma^{(d)} \in \mathcal{G} \mid d \leq d'\}$ | set of simplices of dimension $d'$ or less |

**Local simplex**

| | |
|---|---|
| $B_{t,k}(i) = \{j : i \sim_k j, \ i, j \in \mathcal{G}_{t-}^{(0)}\}$ | $k$-ball centered at vertex $i$ |
| $B_{t,k}(\sigma^{(d)}) = \bigcup_{i:\text{Vert}(\sigma^{(d)})} B_{t,k}(i)$ | $k$-ball centered at simplex $\sigma^{(d)}$ |
| $\mathcal{G}'_t(\sigma^{(d)})$ | all simplices from $\mathcal{G}_t$ spanned by vertices in $B_{t,k}(\sigma^{(d)})$ |

**Feature of simplex**

| | |
|---|---|
| $f_k = f_k(\mathcal{G})$ | total number of $k$-simplices $\sigma^{(k)} \in \mathcal{G}$ |
| $f(\mathcal{G}) = (f_{-1}, f_0, ..., f_{d-1})$ | face-vector of $\mathcal{G}$ |
| $N_t(\sigma^{(d)}) = f(\mathcal{G}'_t(\sigma^{(d)}))$ | feature of $\sigma^{(d)}$ |
| $s(v, v')$ | weighted sum of past co-occurrences of $v, v' \in \sigma^{(d)}$ |
| $h_t(\sigma^{(d)}, v) = \sum_{i=0}^{d} s(v_i, \widetilde{v})$ | scoring function |
| $F_t(\sigma^{(d)}, \widetilde{v}) = (N_t(\sigma^{(d)}), h_t(\sigma^{(d)}, \widetilde{v}))$ | feature vector |
| $P_t(\sigma^{(d)}, F)$ | set of $\sigma^{(d)}, \widetilde{v}$ with corresponding feature $F$ |

$v, v' \in \mathcal{G}_{-}^{(0)}$, we denote by $s(v, v')$ the weighted sum of all past co-occurrences of vertices $v$ and $v'$ in $\sigma^{(d)}$, where the weight is $d$ from $\sigma^{(d)}$. We then devise a scoring function $h(\cdot, \cdot)$ that assigns an integral score to the possible introduction of a vertex $\widetilde{v}$ to a $d$-simplex $\sigma^{(d)} = [v_0, \cdots, v_d]$ as

$$h_t(\sigma^{(d)}, \widetilde{v}) = \sum_{i=0}^{d} s(v_i, \widetilde{v}). \tag{1}$$

Intuitively, the scoring function describes *higher co-occurrence* between $\sigma^{(d)}$ and $\widetilde{v}$ at time $t$, indicates a higher likelihood of forming a $(d + 1)$-simplex $\tau^{(d+1)} = [\sigma^{(d)}, \widetilde{v}]$ together at a future time $t + 1$. We give a higher score to past co-occurrences of vertex pairs in higher dimensional simplices.

**Feature vector:** Finally, for a given $d$-simplex $\sigma^{(d)}$ at time $t$, and a possible introduction of a new vertex $\widetilde{v} \in B_{t,k}(\sigma^{(d)})$, we assign a feature vector

$$F_t(\sigma^{(d)}, \widetilde{v}) = (N_t(\sigma^{(d)}), h_t(\sigma^{(d)}, \widetilde{v})). \tag{2}$$

We denote the set of all such possible $(\sigma^{(d)}, \widetilde{v})$ pairs with their corresponding feature vectors equal to $F$ as $P_t(\sigma^{(d)}, F)$. Furthermore, among the pairs in $P_t(\sigma^{(d)}, F)$, we denote by $P_t^\tau(\sigma^{(d)}, F)$ those set of pairs with feature vectors equal to $F$ that *actually* form $\tau^{(d+1)} = [\sigma^{(d)}, \widetilde{v}]$ at time $t$, i.e., $\sigma^{(d)}$ appears as a face in $\tau^{(d+1)}$ at time $t$. Note the distinction that not all $d$-simplices counted in $P_t(\sigma^{(d)}, F)$ end up being *promoted* to higher $(d + 1)$-simplices in the next time step. Table 1 provides a list of notations.

### 3.2 PREDICTION MODEL AND KERNEL ESTIMATOR

For the prediction, we define an indicator variable that displays the appearance of a new simplex. Given a $d$-simplex $\sigma^{(d)} = [v_0, \cdots, v_d] \in \mathcal{G}_t$, the arrival at time $t + 1$ of a $(d + 1)$-simplex $\tau^{(d+1)} = [v_0, \cdots, v_d, \widetilde{v}]$ with a new vertex $\widetilde{v} \in B_{t,k}(\sigma^{(d)})$ is captured by the following variable

$$Y_{t+1}(\tau^{(d+1)}) := \begin{cases} 1 & \text{if } \sigma^{(d)} \text{ is a face of } \tau^{(d+1)}, \\ 0 & \text{otherwise.} \end{cases} \tag{3}$$

**Prediction model**: Our approach for the prediction is to model the indicator variable. Namely, we assume that the indicator variable follows the following distribution:

$$Y_{t+1}(\tau^{(d+1)}) \mid \mathcal{G}_{t,p} \sim \text{Bernoulli}(g(F_t(\sigma^{(d)}, \widetilde{v}))) \tag{4}$$

where $0 \leq g(\cdot) \leq 1$ is a function of the feature vector $F_t(\sigma^{(d)}, \widetilde{v})$. In words, the indicator variable $Y_{t+1}(\tau^{(d+1)})$ is Bernoulli distributed with success probability given by function $g(F_t(\sigma^{(d)}, \widetilde{v}))$, conditioned on having seen the last $p$ states of GSC $\mathcal{G}_t$. This model describes that the appearance probabilities for two simplices $\sigma_i$ and $\sigma_j$ are likely to be similar, if their feature vector is also similar.

**Estimator with Kernels**: We utilize a kernel method to estimate the success probability of the model (Equation 4) based on observed simplices at time $t$. Let $\mathcal{G}_t(d)$ be a set of $d$-dimensional simplices at time $t$ from $\mathcal{G}_{t,p}$. Also, for brevity, let $F$ represent a feature $F_t(\sigma^{(d)}, \widetilde{v}_m)$ with some $t, \sigma^{(d)}$ and $\widetilde{v}_m$ subject to $\widetilde{v}_m \in B_{t,k}(\sigma_i^{(d)})$. Let $\|F - F'\|_1$ denote the $L_1$-distance between two feature vectors, and also define a $L_1$-ball $\Gamma(F, \delta) := \{F' : \|F - F'\|_1 \leq \delta\}$.

We define our kernel function $K(\cdot, \cdot)$ as follows. With two features $F$ and $F'$, we define it as

$$K(F, F') := \frac{\mathbb{I}\{F = F'\} + \beta \mathbb{I}\{\|F - F'\|_1 \leq \delta\}}{1 + \beta|\Gamma(F, \delta)|}, \tag{5}$$

where $\beta > 0$ is the bandwidth parameter, and $\mathbb{I}$ is the indicator function. Given the feature $F$ with only integer components, we are interested in only those *close by feature vectors* that are either exactly the same as $f$ or lie within an $L_1$-ball of radius $\delta$ centered on $f$. This explains the choice of our kernel function with discrete indicator variables.

Now, we define our estimator. At time $T$, we fix $\sigma^{(d)}$ and $\widetilde{v}$ and set $F = F_T(\sigma^{(d)}, \widetilde{v})$, our estimator of $g(F)$ in (Equation 4) is written as follows:

$$\widetilde{g}_T(F) := \frac{\sum_{t'=T-p}^{T} \sum_{\sigma_j^{(d)} \in \mathcal{G}_{t'}(d)} \sum_{\widetilde{v}_n \in B_{t',k}(\sigma_j^{(d)})} K\left(F, F_{t'}(\sigma_j^{(d)}, \widetilde{v}_n)\right) \cdot Y_{t'+1}([\sigma_j^{(d)}, \widetilde{v}_n])}{\sum_{t'=T-p}^{T} \sum_{\sigma_j^{(d)} \in \mathcal{G}_{t'}(d)} \sum_{\widetilde{v}_n \in B_{t',k}(\sigma_j^{(d)})} K\left(F, F_{t'}(\sigma_j^{(d)}, \widetilde{v}_n)\right)}. \tag{6}$$

**Time-complexity of our estimator**: As both the face-vectors and counts $P_{t+1}$ / $P_{t+1}^{(\tau)}$ (updated in a data cube) are computed simultaneously, they incur the same time overhead. At time $t$, for a $d$-simplex $\sigma^{(d)}$ it takes $O(|V_k| + |E_k|)$ time to compute a $k$-ball around $\sigma^{(d)}$, where $V_k$ and $E_k$ are the set of vertices and edges in the $k$-hop subgraph of a vertex. We must check this $k$-ball of $\sigma^{(d)}$ against $p|\mathcal{G}_{t-}^{(d)}|$ number of simplices (with dimension at most $d$) from the previous $p$ time steps, by intersecting against them to get counts for the face vector and data cube. Each intersection test takes $O(|V_k|)$ time. Recall that $f_d(\mathcal{G}_t)$ denoted the total number of $d$-simplices in $\mathcal{G}_t$. $V_k$ and $E_k$ are the set of vertices and edges in the $k$-hop subgraph of a vertex. We must check this $k$-ball of $\sigma^{(d)}$ against $p|\mathcal{G}_{t-}^{(d)}|$ number of simplices (with dimension at most $d$) from the previous $p$ time steps. So, the entire computation across $T$ time chunks has a time complexity of $O(Tpf_d(\mathcal{G}_t)(|V_k| + |E_k|))|\mathcal{G}_{t-}^{(d)}|)$.

**Storage-complexity of our estimator**: For a single simplex, our estimator requires storing: (i) a pair of integer counts, namely $(P_t(\cdot, \cdot), P_t^\tau(\cdot, \cdot))$, in a datacube, which costs $O(1)$ and (ii) a $d+1$-dimensional face vector which takes storage $O(d+1)$. As the total number of simplices is denoted by $f_d(\mathcal{G}_t)$, we arrive at a total storage cost of $O(df_d(\mathcal{G}_t))$.

## 4 THEORETICAL PROPERTY OF THE ESTIMATOR

We show that our estimator has theoretical validity: (i) consistency and (ii) asymptotic normality. (i) The consistency guarantees $\widetilde{g}_T$ achieves zero error as $T$ increases by converging to $g$. (ii) The asymptotic normality implies that the error $\widetilde{g}_T - g$ converges to a normal distribution, which is useful to evaluate the size of the error and can be applied to statistical tests and confidence analysis. Both properties are very important in statistics (Van der Vaart, 2000).

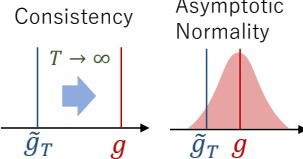

### 4.1 CONSISTENCY

We study the consistency of our estimator. To discuss the property of the estimators with GSCs, it is necessary to organize the *Markov property*, that the GSC evolution process clearly exhibits. It is

well-known that there exists a set of irreducible closed communication classes $C$ in the state space $\mathcal{S}$. We denote the time of entering class $C$ by $T_C$ and the event as $\mathcal{E}(T_C)$. Let $S_C$ denote the event $S_t \in C$, where $S_t$ is the state of the Markov chain at time $t$. Then, $\mathcal{E}(T_C) \cap S_C$ is the event that the chain enters class $C$ at time $T_C$ and remains in that communication class indefinitely.

With the event $S_C$, we provide the bias-variance decomposition of $\widetilde{g}_T - g$, which is common for theoretical analysis of estimators. The bias represents an error due to the expressive power of the model and the variance represents the over-fitting error due to algorithm uncertainty. By analyzing these terms separately, we can analyze the overall prediction error. We define two functions as $\widehat{h}_T(F) = (T-p)^{-1} \sum_{t=p}^{T-1} \sum_{j=1}^{|\mathcal{G}_t(d)|} |\mathcal{G}_t(d)|^{-1} |P_{t+1}^\tau(\sigma_j^{(d)}, F)|$ and $\widehat{d}_T(F) = (T-p)^{-1} \sum_{t=p}^{T-1} \sum_{j=1}^{|\mathcal{G}_t(d)|} |\mathcal{G}_t(d)|^{-1} |P_{t+1}(\sigma_j^{(d)}, F)|$. For the sake of brevity, we fix $F$ and denote $\widetilde{g}_T(F)$ by $\widetilde{g}_T$. Similarly, $g, \widehat{h}_T$ and $\widehat{d}_T$ are used. Also, we define a term $B_T(F, C) = \mathbb{E}[\widehat{h}_T \mid S_C]/\mathbb{E}[\widehat{d}_T \mid S_C] - g$. Then, by Proposition 2 in the supplementary material, we decompose $\widetilde{g}_T - g$ as

$$\widetilde{g}_T - g = \underbrace{\frac{[\widehat{h}_T - g\widehat{d}_T] - \mathbb{E}[\widehat{h}_T - g\widehat{d}_T \mid S_C]}{\widehat{d}_T}}_{=:\mathcal{V}_T \text{ (variance)}} + \underbrace{\frac{B_T(F, C)\mathbb{E}[\widehat{d}_T \mid S_C]}{\widehat{d}_T}}_{=:\mathcal{B}_T \text{ (bias)}} + o(1). \qquad (7)$$

To bound the variance term $\mathcal{V}_T$, we make an assumption that our Markov chain $X_t$ exhibits a $\alpha$-*mixing* property which describes a dependent property of the dynamic process. It is one of the most common and well-used assumptions describing time-dependent processes including dynamic graphs (Sarkar et al., 2014b). Precisely, we present the definition of $\alpha$-*mixing*:

**Definition 5** ($\alpha$-mixing). *A stochastic process $X_t$ is $\alpha$-mixing, if a coefficient $\alpha(r)$, defined as*

$$\alpha(r) = \sup_{|t_1 - t_2| \geq r} \{|\Pr(A \cap B) - \Pr(A)\Pr(B)| : A \in \Sigma(X_{t_1}^-), B \in \Sigma(X_{t_2}^+)\},$$

*satisfies $\alpha(r) \to 0$ as $r \to \infty$. Here, $\Sigma(X_{t_1}^-)$ and $\Sigma(X_{t_2}^+)$ are the sigma-algebras of past and future events of the stochastic process up to and including $t_1$.*

This definition implies that time-dependent processes get *close to independent* as time passes. That is, events at time $t$ and $t + 10000$ are close to independent, while events at time $t$ and $t + 1$ can be correlated. The simplest example is the evolution of stock prices in financial markets: the movement of stock prices today does not correlate with the movement of stock prices 10 years ago.

To bound the bias term $\mathcal{B}_T$, we impose a smoothness condition on $g$. A similar assumption is often used in the problem of predicting links (e.g. Assumption 1 in Sarkar et al. (2014a)). Our assumption is a general and weaker version of the common assumption.

**Assumption 1** (Smoothness on $g$). *There exists a function $\kappa : \mathbb{R} \to \mathbb{R}$ in the Schwartz space (i.e. it is infinitely differentiable and converging to zero faster than any polynomial as $x \to \pm\infty$) with a parameter $b > 0$ such that $|g(F) - g(F')| = O(\kappa(-\|F - F'\|_1/b))$, as $b \to 0$, $\forall F, F'$.*

We utilize all the assumptions, then prove the consistency of $\widetilde{g}_T$.

**Theorem 1** (Consistency). *Suppose that the GSC filtration process is $\alpha$-mixing, $\beta = o(1)$, and Assumption 1 holds. Then, for any $F$ and conditional on $S_C$, our estimator $\widetilde{g}_T(F)$ is well-defined with probability tending to 1, and $|\widetilde{g}_T(F) - g(F)| \xrightarrow{p} 0$ holds as $T \to \infty$.*

### 4.2 ASYMPTOTIC NORMALITY

We show the asymptotic normality of the proposed estimator. That is, we prove that the error of the estimator converges weakly to a normal distribution. This property allows for more detailed investigations, such as correcting for errors in estimators or performing statistical tests.

Technically speaking, we develop a distribution approximation result with *Wasserstein distance* (Villani, 2008) and Stein's method (Stein et al., 1986) to handle the dependency property of GSCs. We are interested in approximating a random variable $Z_n$ by a Gaussian random variable, where $Z_n$ is a sum of $n$ mean-centered random variables $\{A_i\}_{i=0}^n$, where $A_i$ corresponds to a random variable which depends on the $i$-th $d$-simplex $\sigma_i^{(d)}$ in our GSC, which is *dependent* on other $d$-simplices

whose neighborhoods largely overlap with that of $\sigma_i^{(d)}$. Here, let $d_w$ be the Wasserstein distance between the underlying distributions of the random variables, and $N$ is a standard Gaussian variable. Then, we develop the general results for $Z_n$. We provide the following theoretical result. Its formal statement is Theorem 3, which is deferred to the supplementary material due to its complexity.

**Proposition 1** (Gaussian approximation for dependent variables; Simple version of Theorem 3). *Suppose* $Z_n = \sum_{i=0}^n A_i$, *where* $\{A_i\}_{i=0}^n$ *is generated by zero-mean random variables* $X_0, X_1, \cdots, X_n$ *satisfying the $\alpha$-mixing condition, such as* $A_i = X_i / B_n$ *with* $B_n = \mathbb{E}[\sum_{i=0}^n X_i^2]$. *Also, suppose that* $Pr\{|X_i| \leq L\} = 1$ *holds for* $i = 1, \cdots, n$ *with some constant* $L > 0$. *Then, with an existing finite constant* $C > 0$, *we have* $d_w(Z_n, N) \leq C \left( \sum_{i=1}^n \mathbb{E}|A_i|^3 + \frac{TL^3}{B_n^3} \sum_{i=1}^{n-1} n\alpha(n) \right)$.

This result extends Sunklodas (2007) in the sense of Markov chains on GSCs that satisfy the $\alpha$-mixing condition. This extension makes it possible to study the *contributing effect* of neighboring $d$-simplices (represented as a set of weakly dependent r.v.'s) on a central $d$-simplex.

We provide the asymptotic normality of our estimator. It shows that our estimator converges to a normal distribution in terms of the Wasserstein distance, which leads to weak convergence. To achieve the result, we utilize the decomposed terms $\mathcal{V}_T$ and $\mathcal{B}_T$ from (7). Then, we regard $\mathcal{V}_T$ as a sum of dependent random variables and apply the result developed in Proposition 1. Let $\sigma_c^2$ be a limit of variance of the numerator in $T^{-1/2}\mathcal{V}_T$ as $T \to \infty$. We recall that $S_C$ denotes the event $S_t \in C$, where $S_t$ is the state of the Markov chain at time $t$.

**Theorem 2** (Asymptotic Normality). *Suppose that Assumption 1 holds, the GSC filtration process is $\alpha$-mixing, and $\sigma_c > 0$. If $\beta = o(T^{-1/2})$ and $b = o(T^{-1/2})$, then, for any $F$ and conditioned on $S_C$, the following holds:* $\sqrt{T}(\widetilde{g}_T(F) - g(F)) \xrightarrow{d} \mathcal{N}(0, \sigma_c^2/R(C)^2)$, *as* $T \to \infty$.

By using this property, we can make detailed inferences based on the distribution of the estimation error. For example, it is possible to create confidence intervals for predictions and perform statistical tests to rigorously test hypotheses about simplex arrivals.

## 5 REAL-WORLD DATA EXPERIMENTS

We empirically evaluate the performance of our proposed estimator on real-world dynamic graphs compared to baselines. The basic premise in our experiments is to capture local and higher-order properties surrounding a $d$-simplex up to time $t$ to predict the appearance of a new $(d+1)$-simplex at time $t' > t$, which contains $\sigma^{(d)}$ as its *face*. Note that we compare our method to other *closely related* methods that were designed to solve different structure prediction tasks.

**Datasets**: We report results on real-world dynamic graph datasets sourced from Benson et al. (2018). Each dataset contains $n$ nodes, $m$ formed edges, and $x$ *timestamped simplices* (represented as a set of nodes). There are four datasets named *Enron* ($n = 143$, $m = 1.8$K, $x = 5$K), *EU* ($n = 998$, $m = 29.3$K, $x = 8$K), *Contact* ($n = 327$, $m = 5.8$K, $x = 10$K), and *NDC (National Drug Code)* ($n = 1.1$K, $m = 6.2$K, $x = 12$K).

**Experimental setup**: We first ordered by arrival times and grouped the timestamped simplices into $T$ time slices. For most of our experiments, $T$ was set to 20, except for $d = 2$, where $T$ was set to 6 and 12 for *EU* and *NDC*, respectively. Then, we randomly sampled a set of $d$-simplices from the time slices in the range $[1, T-1]$. Those $d$-simplices paired with a vertex that successfully formed a face in a $(d+1)$-simplex in the $T$-th time slice were classified as *positive samples*, while the rest were deemed as *negative samples*. We picked an equal number of positive and negative samples for evaluation. For $K$-fold cross-validation for $\beta$, we swapped the $T$-th time slice with one of the $K$ slices preceding the $T$-th time slice for each fold. $K$ was set to 3. All experiments where repeated 10 times and average AUC scores and runtimes are reported.

**Compared methods**: As naive baselines, we averaged the results of single-edge prediction methods, where a new edge would form between each node in the $d$-simplex and the vertex to be paired with. Specifically, we compare our estimator with: (i) *heuristic* (*Adamic-Adar* (AA) (Adamic & Adar, 2001), *Jaccard Coefficient* (JC) (Salton & McGill, 1986), and *Preferential attachment* (PA) (Mitzenmacher, 2004), (ii) *deep-learning* based (*Node2vec* (NV) (Grover & Leskovec, 2016a), and *SEAL* (SL) (Zhang & Chen, 2018a)), and (iii) *temporal graph network* based (*TGAT* (TT) da Xu et al.

Table 2: AUC scores and runtimes for baselines versus our method's estimator for $d = 1$ and $d = 2$.

|  | Enron | Contact | NDC | EU | Enron | Contact | NDC | EU |
|---|---|---|---|---|---|---|---|---|
|  | ($d = 1$) **AUC / runtime (sec)** | | | | ($d = 2$) **AUC / runtime (sec)** | | | |
| AA | 0.54 / 0.18 | 0.57 / 0.18 | 0.30 / 0.38 | 0.61 / 0.15 | 0.31 / 0.16 | 0.33 / 0.20 | 0.47 / 0.25 | 0.25 / 0.28 |
| JC | 0.42 / 0.20 | 0.63 / 0.2 | 0.16 / 0.37 | 0.65 / 0.20 | 0.41 / 0.16 | 0.44 / 0.21 | 0.23 / 0.24 | 0.32 / 0.27 |
| PA | 0.55 / 0.15 | 0.60 / 0.28 | 0.55 / 0.11 | 0.38 / 0.09 | 0.52 / 0.15 | 0.63 / 0.20 | 0.74 / 0.24 | 0.34 / 0.18 |
| NV | 0.45 / 74 | 0.25 / 155 | 0.30 / 406 | 0.67 / 280 | 0.49 / 71 | 0.45 / 155 | 0.49 / 3989 | 0.40 / 374 |
| SL | 0.54 / 152 | **0.91** / 241 | 0.33 / 260 | 0.57 / 431 | 0.48 / 152 | 0.54 / 241 | 0.40 / 260 | 0.29 / 431 |
| TT | 0.62 / 180 | 0.80 / 202 | 0.42 / 285 | 0.61 / 420 | 0.70 / 180 | 0.78 / 205 | 0.45 / 285 | 0.67 / 422 |
| TN | 0.67 / 212 | 0.84 / 241 | 0.48 / 280 | 0.55 / 512 | 0.62 / 212 | 0.70 / 241 | 0.46 / 281 | 0.64 / 512 |
| HP | 0.26 / 22 | 0.64 / 142 | 0.57 / 54 | 0.45 / 17 | 0.26 / 22 | 0.76 / 144 | 0.45 / 58 | 0.41 / 18 |
| Ours | **0.88** / 1.08 | 0.87 / 7.56 | **0.78** / 5.84 | **0.83** / 1.48 | **0.94** / 2.25 | **0.83** / 2.76 | **0.96** / 0.76 | **0.80** / 0.642 |
|  | ($\beta$=1) | ($\beta$=0.1) | ($\beta$=0.1) | ($\beta$=10) | ($\beta$=0.01) | ($\beta$=0.01) | ($\beta$=0.01) | ($\beta$=10) |

(2020) and *TGN* (TN) Rossi et al. (2020)) link prediction methods. We note that (Benson et al., 2018) for predicting a "simplicial closure" has the closest motivation to our method, yet has divergent objectives, therefore we omit comparison to their work. For *hyper-edge prediction* (HP), we picked the recent most representative work by Yoon et al. (2020) to compare against, although this work only works for static non-evolving hypergraphs.

## 5.1 RESULTS AND DISCUSSION

We averaged the classification accuracy and runtimes of our estimator and the baselines. We performed two sets of experiments on the arrival of a $(d + 1)$-simplex and summarize it in Table 2 for $d = \{1, 2\}$. We also report the bandwidth $\beta$ for our estimator selected by cross-validation.

**Predicting** 2**-simplex** ($d = 1$): We observe that our method is nearly two orders of magnitude faster than the deep learning based methods (NV and SL) and nearly an order of magnitude faster than the hypergraph prediction method (HP). While the single edge heuristic methods are relatively faster, their AUC scores are not comparable to our method's AUC scores. Also, we achieve nearly 30% improvement (in Enron) over the next best performing prediction method.

**Predicting** 3**-simplex** ($d = 2$): The gap in AUC scores between our method and the baselines are far more pronounced. Our runtimes also improve due to the far fewer number of simplices with dimensions exceeding 3. As observed in Yoon et al. (2020) about slight drops in accuracy for higher-dimensional hyper-edges, we also note that in HP, the AUC score remains the same or drops slightly compared to prediction at $d = 1$.

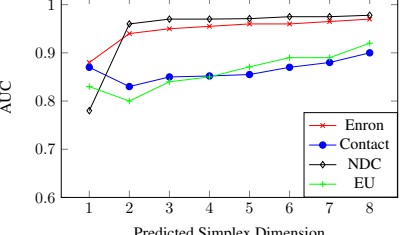

Figure 3: AUC score predicted by our estimator for future formation of a $d$-simplex, given a $d - 1$-simplex.

**Advantage of higher dimensional simplices**: We perform additional experiments by increasing $d$ from 1 to 8 and show that handling high-dimensional simplices exhibits high prediction accuracy. In Figure 3, the prediction is basically improves as $d$ increases.

**Empirical summary**: Traditional estimators fail to accurately capture the rich latent information present in higher-order structures (and their sub-structures) that evolve over time. Our estimator succinctly captures this information via the $f$-vector and weighted scoring of $(\sigma^{(d)}, v)$ pair formation depending on the dimension of the simplex in which the pair co-occur in the past.

## 6 CONCLUSION

We modeled the higher-order interaction as a *simplex* and demonstrated a novel *kernel estimator* to solve the higher-order structure prediction problem. From a theoretical standpoint, we proved the consistency and asymptotic normality of our estimator. We empirically argue that our estimator outperforms hypergraph based and higher-order link prediction baselines from both heuristic and deep-learning based pairwise link prediction methods.

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

# Supplementary Materials of Higher-order Link Prediction in Dynamic Graph Simplicial Complexes

## A   EXAMPLE OF SIMPLEX AND RELATED NOTIONS

**Example 1.** We begin by computing the $k$-balls centered at 1-simplex $[9, 10]$ in $\mathcal{G}_t$ and $\mathcal{G}_{t-1}$, respectively.

The $k$-ball at time $t$ for $k = 1$ (i.e., 1-hop vertices only) centered at $[9, 10]$ is:

$B_{t,1}([9, 10]) = B_{t,1}([9]) \cup B_{t,1}([10])$. This is the union of $k$-balls at underlying vertices 9 and 10 according to Definition 3.

$$
\begin{aligned}
B_{t,1}([9, 10]) &= B_{t,1}([9]) \cup B_{t,1}([10]) \\
&= \{9, 13, 8, 5, 6, 10\} \cup \{10, 14, 13, 9, 6, 7, 11, 15\} \\
&= \{9, 13, 8, 5, 6, 10, 14, 7, 11, 15\}
\end{aligned}
$$

Similarly, The $k$-ball at previous time step $t - 1$ for $k = 1$ (i.e., 1-hop vertices only) centered at $[9, 10]$ is:

$$
\begin{aligned}
B_{t-1,1}([9, 10]) &= B_{t-1,1}([9]) \cup B_{t-1,1}([10]) \\
&= \{9, 13, 8, 5, 6, 10\} \cup \{10, 14, 13, 9, 6, 11, 15\} \\
&= \{9, 13, 8, 5, 6, 10, 14, 11, 15\}
\end{aligned}
$$

Notice that there is only a difference of vertex 7 missing from set $B_{t-1,1}([9, 10])$ as compared to set $B_{t,1}([9, 10])$ at time $t$.

Now, we calculate the subcomplex spanned by $B_{t,1}([9, 10])$ Then[3],

$$
\begin{aligned}
\mathcal{G}'_t([9, 10]) =\{&[9], [13], [8], [5], [6], [10], [14], [11], [15], [7], \qquad\qquad\qquad (8)\\
&[9, 10], [9, 6], [9, 5], [9, 8], [9, 13], [10, 6], [10, 7], [10, 11], [10, 15], [10, 14], [10, 13], \\
&[9, 6, 10], [10, 14, 15]\}
\end{aligned}
$$

Now, we compute the compressed $f$-vector notation of $\mathcal{G}'_t([9, 10])$ to get

$$
f(\mathcal{G}'_t([9, 10])) = (1, 10, 11, 2)
$$

Finally, the neighborhood $N_t([9, 10]) = (1, 10, 11, 2)$.

## B   FURTHER DETAILS OF THE EXPERIMENT

### B.1   DATASETS

We report results on real-world dynamic graph datasets sourced from Benson et. al. Benson et al. (2018). Each dataset is a set of *timestamped simplices* (represented as a set of nodes). In each dataset, let $n$, $m$, and $x$ denote the number of nodes, edges formed, and timestamped simplices, respectively. *Enron* ($n = 143$, $m = 1.8$K, $x = 5$K) and *EU* ($n = 998$, $m = 29.3$K, $x = 8$K) model email networks where nodes are email addresses and all recipients of an email form a simplex in the network. *Contact* ($n = 327$, $m = 5.8$K, $x = 10$K) is a proximity graph where nodes represent persons and a simplex is a set of persons in close proximity to each other. *NDC* ($n = 1.1$K, $m = 6.2$K, $x = 12$K) is a drug network from the *National Drug Code* directory, where nodes are class labels and a simplex is formed when a set of class labels appear together on a single drug.

### B.2   COMPARED METHODS

*Adamic-Adar* Adamic & Adar (2001) and the *Jaccard Coefficient* Salton & McGill (1986) measure link probability between two nodes based on the closeness of their respective feature vectors. *Preferential attachment* Mitzenmacher (2004) has received considerable attention as a model of growth of

---

[3]all simplices are placed on a line each in increasing order of their dimension

networks as they model future link probability as the product of the current number of neighbors of the two nodes. Motivated by resource allocation in transportation networks (much alike the Optimal Transport (OT) problem), *Resource allocation index* Zhou et al. (2009b) proposes a node $x$ tries to transmit a unit resource to node $y$ via common neighbors that play the role of *transmitters* and similarity is measured by the amount of the resource $y$ received from $x$. Node2vec Grover & Leskovec (2016a) and SEAL Zhang & Chen (2018a) are deep-learning based graph embedding methods that are used in link prediction.

**Remark 1** (Difference between our setting and *simplicial closure*). The closest work Benson et al. (2018) proposed predicting a "simplicial closure" event where at time $t$ there exists a set of nodes which are pairwise edge connected and the task is to predict whether at time $t + 1$ there will arrive a simplex which covers all these nodes. This phenomenon was termed as *simplicial closure*. For example, authors $A$, $B$ and $C$ have all co-authored in pairs (i.e., $\{A, B\}$, $\{A, C\}$ and $\{B, C\}$) and a simplicial closure event would take place at $t + 1$, if a simplex $\{A, B, C\}$ arrives, implying that all three authors co-author on a single paper. Our prediction task significantly diverges and aims to solve a different problem. Considering our previous example, we are given a single co-authorship relationship between say $A$ and $B$ at time $t$, we predict whether authors $A$ and $B$ will co-author with a third author $C$ (ternary co-authorship relationship) on a single paper at time $t + 1$ in the future.

## C  PROOF FOR CONSISTENCY

For preparation, we rewrite the estimator $\widehat{g}_T$. Plugging in the definition of our kernel (Equation 5) into the equation of our estimator (Equation 6) along with the definitions of $P_t(\cdot, \cdot)$ and $P_t^\tau(\cdot, \cdot)$ to replace the indicator variables with actual counts, we obtain the following simplification of Equation 6. Then, it is reformulated as

$$\widetilde{g}_T(F) = \frac{\sum_{t'=T-p}^{T} \sum_{\sigma_j^{(d)} \in \mathcal{G}_{t'}(d)} \left( |P_{t'+1}^\tau(\sigma_j^{(d)}, F)| + \beta \sum_{s \in \Gamma(F,\delta)} |P_{t'+1}^\tau(\sigma_j^{(d)}, s)| \right)}{\sum_{t'=T-p}^{T} \sum_{\sigma_j^{(d)} \in \mathcal{G}_{t'}(d)} \left( |P_{t'+1}(\sigma_j^{(d)}, F)| + \beta \sum_{s \in \Gamma(F,\delta)} |P_{t'+1}(\sigma_j^{(d)}, s)| \right)}. \quad (9)$$

When we set $\beta = 0$, we look for other pairs whose feature corresponds to $F$ and we calculate a fraction of how many such close by pairs actually form a $(d + 1)$-simplex at time $t' + 1$. This fraction is summed across various $d$-simplices $\sigma_j^{(d)}$ by varying $j$ and also across various discrete time steps by varying $t'$. Setting $\beta > 0$ allows our estimator to *smooth* over close by features.

It turns out simpler to study a proxy estimator $\widehat{g}_T$, which omits the smoothing of the original one. We will show that it is asymptotically equivalent to $\widetilde{g}$. Let $|\mathcal{G}_t(d)|$ denote the total number of $d$-simplices in GSC $\mathcal{G}_t$ at time $t$. Then, for a feature $F$, we define the proxy estimator as

$$\widehat{g}_T(F) := \frac{\widehat{h}_T(F)}{\widehat{d}_T(F)}, \quad (10)$$

where the terms are defined as

$$\widehat{h}_T(F) = \frac{1}{T-p} \sum_{t=p}^{T-1} \sum_{j=1}^{|\mathcal{G}_t(d)|} \frac{|P_{t+1}^\tau(\sigma_j^{(d)}, F)|}{|\mathcal{G}_t(d)|}, \text{ and } \widehat{d}_T(F) = \frac{1}{T-p} \sum_{t=p}^{T-1} \sum_{j=1}^{|\mathcal{G}_t(d)|} \frac{|P_{t+1}(\sigma_j^{(d)}, F)|}{|\mathcal{G}_t(d)|}.$$

Observe that $\sum_{i=1}^{|\mathcal{G}_t(d)|}$ ranges over the total number of $d$-simplices in $\mathcal{G}_t$[4] and note that $|\mathcal{G}_t(d)|$ changes with time step $t$. Recall that the terms $P_{t+1}^\tau(\cdot, \cdot)$ and $P_{t+1}(\cdot, \cdot)$ count all *actual* and *possible* formation of $[\sigma_i^{(d)}, \widetilde{v}_m]$ $((d + 1)$-simplex) that have the *same feature vector* $F$. Lemma 1 in the supplementary material proves $|\widetilde{g}_T(F) - \widehat{g}_T(F)| \to 0$ as $\beta \to 0$. First of all, we show the validity of the proxy estimator $\widehat{g}_T$.

**Lemma 1** (Approximation by proxy). *We obtain*

$$|\widetilde{g}_T(F) - \widehat{g}_T(F)| = O(\beta), \forall F.$$

---

[4]In practice, the total number of $d$-simplices is much less than the maximum possible cliques with $n$ vertices.

*Proof of Lemma 1.* Recall that $\Gamma(F, \delta)$ denotes the set of features at $L_1$-distance at most $\delta$ from $F$. We denote by $|\Gamma(F, \delta)|$ the cardinality of this set.

$$\widetilde{g}_T(F) = \frac{\widehat{h}_T(F) + C_1}{\widehat{d}_T(F) + C_2}$$

where $C_1 = \beta \sum_{j,t} \sum_{s \in \Gamma(F,\delta)} |P_{t+1}^\tau(\sigma_j^{(d)}, s)|$ and $C_2 = \beta \sum_{j,t} \sum_{s \in \Gamma(F,\delta)} |P_{t+1}(\sigma_j^{(d)}, s)|$. Due to the finiteness of features in $P_{t+1}^\tau(\cdot, \cdot)$ and $P_{t+1}(\cdot, \cdot)$, we have that $C_1 = C_2 = O(\beta)$. Both $C_1$ and $C_2$ are non-negative integers. So,

$$|\widetilde{g}_T(F) - \widehat{g}_T(F)| = \left| \frac{\widehat{h}_T(F) + C_1}{\widehat{d}_T(F) + C_2} - \frac{\widehat{h}_T(F)}{\widehat{d}_T(F)} \right| = O(\beta)$$

In the last step, the second fraction is a positive constant and can thus be ignored from our asymptotic analysis because both $\widehat{h}_T$ and $\widehat{d}_T$ are bounded. $\qquad\square$

Next, we prove the convergence of the proxy estimator $\widehat{g}_T(F)$. As the first step, we describe the detail of its decomposition in (7). To simplify the notation, we will drop $F$ from all estimator notations.

**Proposition 2.** *As written in Equation 7, we obtain*

$$\widehat{g}_T - g = \mathcal{V}_T + \mathcal{B}_T.$$

*Furthermore, with the event $S_C$, there exist a stochastic terms $q_t$ for $t$ such as*

$$\mathcal{V}_T = \frac{(T-p)^{-1} \sum_{t=p}^{T-1} q_t}{\widehat{d}_T}.$$

*Proof of Proposition 2.* With the definition of $B_T(F, C)$, we have

$$\widehat{g}_T - g = \frac{\widehat{h}_T}{\widehat{d}_T} - g \tag{11}$$

$$= \frac{\widehat{h}_T - g\widehat{d}_T}{\widehat{d}_T}$$

$$= \frac{[\widehat{h}_T - g\widehat{d}_T] - \mathbb{E}[\widehat{h}_T - g\widehat{d}_T \mid S_C] + \mathbb{E}[\widehat{h}_T - g\widehat{d}_T \mid S_C]}{\widehat{d}_T}$$

$$= \frac{[\widehat{h}_T - g\widehat{d}_T] - \mathbb{E}[\widehat{h}_T - g\widehat{d}_T \mid S_C]}{\widehat{d}_T} + \frac{\mathbb{E}[\widehat{h}_T \mid S_C] - g\mathbb{E}[\widehat{d}_T \mid S_C]}{\widehat{d}_T}$$

$$= \frac{[\widehat{h}_T - g\widehat{d}_T] - \mathbb{E}[\widehat{h}_T - g\widehat{d}_T \mid S_C]}{\widehat{d}_T} + \frac{B_T(F, C)\mathbb{E}[\widehat{d}_T \mid S_C]}{\widehat{d}_T}$$

$$= \mathcal{V}_T + \mathcal{B}_T. \tag{12}$$

We are interested in the asymptotic behavior of the Markov chain at time $T \to \infty$. Let $F$ denote $F_T(\sigma_i^{(d)}, \widetilde{v}_m)$. Recall that the terms $|P_{T+1}^\tau(\cdot, \cdot)|$ and $|P_{T+1}(\cdot, \cdot)|$ count all *actual* and *possible* formation of $[\sigma_i^{(d)}, \widetilde{v}_m]$ $((d+1)$-simplex) that result in the *same feature vector $F$* at time $T+1$. Our Markov chain has a *finite state space* and hence belongs to a closed communication class with probability approaching to 1. We provide a statistical consistency conditional on $S_C$ for any communication class $C$.

For a given time step $t$, we define

$$\widehat{h}_T(t) := \frac{1}{|\mathcal{G}_t(d)|} \sum_{j=1}^{|\mathcal{G}_t(d)|} |P_{t+1}^\tau(\sigma_j^{(d)}, F)| \tag{13}$$

$$\widehat{d}_T(t) := \frac{1}{|\mathcal{G}_t(d)|} \sum_{j=1}^{|\mathcal{G}_t(d)|} |P_{t+1}(\sigma_j^{(d)}, F)|$$

Note that $\widehat{h}_T = \frac{1}{T-p} \sum_{t=p}^{T-1} \widehat{h}_T(t)$ and $\widehat{d}_T = \frac{1}{T-p} \sum_{t=p}^{T-1} \widehat{d}_T(t)$.

Let us set

$$q_t := [\widehat{h}_T(t) - g\widehat{d}_T(t)] - \mathbb{E}[\widehat{h}_T(t) - g\widehat{d}_T(t) \mid S_C] \tag{14}$$

Note that $q_t$ is the numerator of the *stochastic term* in Equation 7 and a *bounded deterministic function* of $S_C$ at a given time step $t$. □

For the stochastic term $\widehat{d}_T$ which appears in the denominator, we show its convergence. The following two lemmas provide the result.

**Lemma 2.** *If the GSC process is $\alpha$-mixing, then, as $T \to \infty$, we obtain*

$$\mathrm{Var}(\widehat{h}_T(F)|S_C) \to 0, \ and \ \mathrm{Var}(\widehat{d}_T(F)|S_C) \to 0,$$

*for any $F$.*

*Proof of Lemma 2.* We show that variance divided by $T$ converges to a non-negative constant. Let $U_T := \sum_t q_t/\sqrt{T}$, where $q_t$ (as shown in Equation 14) is a bounded deterministic function of the state of $X_t$ at time $t$. As demonstrated in Sarkar et. al. Sarkar et al. (2014a), we too break our weighted sum $U_T$ across three time intervals: (i) $[1, T_C - 1]$, (ii) $[T_C, T_C + M - 1]$, and (iii) $[T_C + M, T]$, where $M$ is a constant. Now, from Sarkar et al. (2014a), we simply apply Lemma 5.7 to get that $\mathbb{E}[\mathrm{Var}(U_T \mid \mathcal{E}(T_C), S_C) \mid S_C] \to \sigma_c$, for some $\sigma_c \geq 0$ and from Lemma 5.8, we have that $\mathrm{Var}(\mathbb{E}[U_T \mid \mathcal{E}(T_C), S_C] \mid S_C) = o(1)$.

Now, since *the law of total variance* provides

$$\mathrm{Var}(U_T \mid S_C) = \mathbb{E}[\mathrm{Var}(U_T \mid \mathcal{E}(T_C), S_C) \mid S_C] + \mathrm{Var}(\mathbb{E}[U_T \mid \mathcal{E}(T_C), S_C] \mid S_C)$$

we use the previous results from Lemmas 5.7 and 5.8 in Sarkar et al. (2014a) to get

$$\mathrm{Var}(U_T \mid S_C) \to \sigma_c \text{ as } T \to 0, \text{ for some constant } \sigma_c \geq 0$$

Plugging in the definition of $q_t$ into $U_T$ and calculating $\mathrm{Var}(U_T \mid S_C)$, it follows trivially that $\mathrm{Var}(\widehat{h}_T \mid S_C) \to 0$ and $\mathrm{Var}(\widehat{d}_T \mid S_C) \to 0$ as $T \to \infty$. We refer readers to Remark 5.10 in Sarkar et al. (2014a) to see how these results also hold in the case when $C$ is *aperiodic*. □

**Lemma 3.** *If the GSC process is $\alpha$-mixing, then there exist a function $R(C)$ with a deterministic function of class $C$ denote, such as*

$$\lim_{T \to \infty} \mathbb{E}[\widehat{d}_T(F) \mid \mathcal{E}(T_C), S_C] = R(C), \ and \ \lim_{T \to \infty} \mathbb{E}[\widehat{d}_T(F) \mid S_C] = R(C).$$

*Proof of Lemma 3.* We know by definition that

$$\mathbb{E}[\widehat{d}_T(F) \mid \mathcal{E}(T_C), S_C] = \frac{1}{T-p} \sum_{t=p}^{T-1} \sum_{j=1}^{|\mathcal{G}_t(d)|} \mathbb{E}\left[\frac{|P_{t+1}(\sigma_j^{(d)}, F)|}{|\mathcal{G}_t(d)|} \ \middle| \ \mathcal{E}(T_C), S_C\right] \tag{15}$$

This is an average of terms $\mathbb{E}\left[\frac{|P_{t+1}(\sigma_j^{(d)}, F)|}{|\mathcal{G}_t(d)|} \ \middle| \ \mathcal{E}(T_C), S_C\right]$ spanning across $d$-simplices with indices $j \in \{1, \cdots, |\mathcal{G}_t(d)|\}$ and discrete time steps $t \in \{p, \cdots, T-1\}$.

For ease of notation, let

$$X_j := \frac{|P_{t+1}(\sigma_j^{(d)}, F)|}{|\mathcal{G}_t(d)|}$$

$X_j$ denotes the total number of possible $(d+1)$-simplices with a $d$-face as $\sigma_j^{(d)}$ divided by the total number of $d$-simplices in $\mathcal{G}_t$.

In the R.H.S. of Equation 15, the term inside the summation is simplified as

$$\mathbb{E}[X_j \mid \mathcal{E}(T_C), S_C] = \sum_x x \Pr[X_j = x \mid \mathcal{E}(T_C), S_C]$$

We know that both $P_{t+1}(\sigma_j^{(d)}, F)$ and $\mathcal{G}_t(d)$ are fully determined given the current state $S_t$ of the Markov chain. Let $\mathbb{I}_S(Y)$ denote an indicator variable of whether "$Y$ is in state $S$" or not.

We have,

$$\Pr[X_j = x \mid \mathcal{E}(T_C), S_C] = \sum_S \mathbb{I}_S(X_j = x) \Pr[S_t = S \mid \mathcal{E}(T_C), S_C]$$

As a result, the R.H.S. of Equation 15 becomes

$$\frac{1}{T} \sum_t \sum_S (\sum_{j,x} x \mathbb{I}_S(X_j = x)) \Pr[S_t = S \mid \mathcal{E}(T_C), S_C] \tag{16}$$

Let $\lambda(S) = \sum_{j,x} x \mathbb{I}_S(X_j = x)$ as this term is fully determined by state $S$. Then, Equation 16 can be rewritten as

$$\sum_S \lambda(S) \frac{\sum_t \Pr[S_t = S \mid \mathcal{E}(T_C), S_C]}{T} \tag{17}$$

Due to stationarity, the average $\sum_t \Pr[S_t = S \mid \mathcal{E}(T_C), S_C]/T$ will converge to a constant function of state $S$, denoted by $R(S)$. Given that $\lambda(S)$ is bounded and the average term converges to a constant $R(S)$, we say that Equation 17 converges to some constant $R(C) > 0$, where $R(C)$ is a deterministic function of communication class $C$. This proves the first part.

A simple application of the *tower property of expectation* followed by the *dominated convergence theorem* shows that $\lim_{T \to \infty} \mathbb{E}[\widehat{d}(F) \mid S_C] = R_C$. This completes the proof for the result. $\square$

Then, we are ready to prove the convergence of the variance term.

**Proposition 3** (Variance). *If the GSC filtration process is $\alpha$-mixing, then, conditional on $S_C$, we obtain $\mathcal{V}_T \xrightarrow{p} 0$ as $T \to \infty$.*

*Proof of Proposition 3.* By Proposition 2, the term $\mathcal{V}_T$ is written as $\frac{(T-p)^{-1} \sum_t q_t}{\widehat{d}_T}$.

For the denominator, Lemma 3 shows that $\mathbb{E}[\widehat{d}_T \mid S_C] \to R(C)$, where $R(C)$ is a positive deterministic function of class $C$. Also, Lemma 2 states that $Var(\widehat{d}_T \mid S_C) \to 0$ holds as $T \to \infty$. Thus, $\widehat{d}_T \xrightarrow{p} R(C) > 0$ holds. Also, $\mathcal{V}_T$ is asymptotically well defined for class $C$.

For the nominator, Lemma 2 also shows that $\lim_{T \to \infty} Var(\sum_t q_t/T \mid S_C) = 0$ as $T \to \infty$ and $\mathbb{E}[q_t \mid S_C] = 0$, therefore we have $\lim_{T \to \infty} 1/T \sum_t q_t \xrightarrow{qm} 0$ conditioned on $S_C$.

By the continuous mapping theorem, we obtain the statement. $\square$

Next, we discuss the bias term $\mathcal{B}_T$. To this aim, we rewrite the term $B_T(F, C)$ as follows:

$$B_T(F, C) = \frac{(\mathbb{E}[\widehat{h}_T(F) \mid S_C] - g(F)\mathbb{E}[\widehat{d}_T(F) \mid S_C])}{\mathbb{E}[\widehat{d}_T(F) \mid S_C]}.$$

**Lemma 4.** *Given that Assumption 1 holds and that when $T \to \infty$, the bandwidth parameter $b \to 0$. Then, we have that $B_T(F, C) = O(b) = o(1)$ as $T \to \infty$.*

*Proof of Lemma 4.* For $t \in [p, T-1]$, $j \in [1, |\mathcal{G}_t(d)|]$ and a fixed feature vector $F$, the numerator of $B_T(F, C)$ can be expressed as an average of terms of the form

$$A_t := \mathbb{E}\left[\frac{|P_{t+1}^\tau(\sigma_j^{(d)}, F)|}{|\mathcal{G}_t(d)|} \,\Big|\, S_C\right] - \mathbb{E}\left[\frac{|P_{t+1}(\sigma_j^{(d)}, F)|}{|\mathcal{G}_t(d)|} \,\Big|\, S_C\right] g(F) \tag{18}$$

The first term in Equation 18 can be rewritten using the *tower property* as

$$\mathbb{E}\left[\mathbb{E}\left[\frac{|P_{t+1}^\tau(\sigma_j^{(d)}, F)|}{|\mathcal{G}_t(d)|} \;\middle|\; \mathcal{E}(T_C), S_C\right] \;\middle|\; S_C\right]$$

When we condition on $\mathcal{E}(T_C)$, it makes $\frac{|P_{t+1}^\tau(\sigma_j^{(d)}, F)|}{|\mathcal{G}_t(d)|}$ conditionally independent of $S_C$, if $t > T_C$. Also, for $t \geq T_C$, we have

$$\mathbb{E}\left[\frac{|P_{t+1}^\tau(\sigma_j^{(d)}, F)|}{|\mathcal{G}_t(d)|} \;\middle|\; \mathcal{E}(T_C), S_C\right] = \frac{|P_{t+1}(\sigma_j^{(d)}, F)|}{|\mathcal{G}_t(d)|} g(F_t(\sigma_j^{(d)}, \widetilde{v}_n)) \tag{19}$$

where $\widetilde{v}_n \in B_{t-1,k}(\sigma_j^{(d)})$.

Given the result in Equation 19 and the fact that the term $\frac{|P_{t+1}^\tau(\sigma_j^{(d)}, F)|}{|\mathcal{G}_t(d)|}$ is bounded results in

$$\mathbb{E}\left[\frac{|P_{t+1}^\tau(\sigma_j^{(d)}, F)|}{|\mathcal{G}_t(d)|} \;\middle|\; \mathcal{E}(T_C), S_C\right] \tag{20}$$

$$\leq \frac{|P_{t+1}(\sigma_j^{(d)}, F)|}{|\mathcal{G}_t(d)|} g(F_t(\sigma_j^{(d)}, \widetilde{v}_n))\mathbb{I}[T_C \leq t] + c\mathbb{I}[T_C > t]$$

$$\leq \frac{|P_{t+1}(\sigma_j^{(d)}, F)|}{|\mathcal{G}_t(d)|} g(F_t(\sigma_j^{(d)}, \widetilde{v}_n)) + c\mathbb{I}[T_C > t],$$

where $c > 0$ an existing constant.

Now, the numerator of $B_T(F, C)$ can be upper bounded as

$$\left|\sum_t A_t/T\right| \leq \sum_t \frac{1}{T} \left|\mathbb{E}\left[\frac{|P_{t+1}(\sigma_j^{(d)}, F)|}{|\mathcal{G}_t(d)|}(g(F_t(\sigma_j^{(d)}, \widetilde{v}_n)) - g(F)) \;\middle|\; S_C\right]\right| \tag{21}$$

$$+ c'\sum_t \Pr[T_C > t]/T.$$

The second term in Equation 21 vanishes as $T \to \infty$ because it is of order $O(\mathbb{E}[T_C]/T)$. Thus, the numerator of $B_T(F, C)$ is an average of terms of the form

$$\mathbb{E}\left[\frac{|P_{t+1}(\sigma_j^{(d)}, F)|}{|\mathcal{G}_t(d)|}(g(F_t(\sigma_j^{(d)}, \widetilde{v}_n)) - g(F)) \;\middle|\; S_C\right]. \tag{22}$$

Our feature vector counts and simplex neighborhoods are finite because $|\mathcal{G}_t(d)|$ is bounded. The expectation in Equation 22 is just a summation of finite terms. We set $F' = F_t(\sigma_j^{(d)}, \widetilde{v}_n)$, and make use of our smoothness assumption 1, so that

$$|g(F') - g(F)| = O(\kappa(-\|F - F'\|_1/b)).$$

We also use Lemma 3 to say that the denominator of our bias term converges to a constant $R(C)$. So,

$$B_T(F, C) = O(\kappa(-\|F - F'\|_1/b)) = O(b).$$

The last equality follows the property of $\kappa$ in the Schwartz space. Now, since $B_T(F, C) = O(b)$ and $b \to 0$ as $T \to \infty$, then we have that $B_T(F, C) = o(1)$. This completes the proof. □

Then, we prove the convergence of the bias. Then, we have the result:

**Proposition 4** (Bias). *If Assumption 1 holds, then, conditional on $S_C$, $\mathcal{B}_T \overset{p}{\to} 0$ holds as $T \to \infty$.*

*Proof of Proposition 4.* Proposition 2 shows that $\mathcal{B}_T(F) = B_T(F, C)/\widehat{d}_T(F)$.

By Lemma 2 and 3, we obtain $\widehat{d}_T \to R(C) > 0$ as $T \to \infty$, as similarly shown in the proof of Proposition 3. Further, Lemma 4 states that $B_T(F, C) = o(b)$. Combining the results, we obtain the statement. □

Now, we can prove the consistency (Theorem 1).

*Proof of Theorem 1.* For the result of $\widehat{g}_T$, we apply the results of Proposition 3 and 4 to the decomposition in the equation 7, then obtain the statement.

For the result of $\widetilde{g}_T$, we additionally combine the result of Lemma 1, then obtain the statement. $\quad\square$

## D   PROOF FOR ASYMPTOTIC NORMALITY

### D.1   INTRODUCTION TO WASSERSTEIN DISTANCE AND APPROXIMATION TECHNIQUE

We denote by $BL(\mathbb{R})$ the space of such bounded functions $h$ that are 1-Lipschitz. More formally,

$$\|h\|_\infty = \sup_{x\in\mathbb{R}}|h(x)| < \infty \text{ and } Lip(h) = 1, \text{ where } Lip(h) = \sup_{x\neq y}\frac{|h(x)-h(y)|}{|x-y|}$$

So, $h \in BL(\mathbb{R})$.

We now make use of the Wasserstein metric to measure the distance between distributions. Therefore, our estimator represented as $W$ can be shown to converge to $Z$, when the Wasserstein distance between $W$ and $Z$'s underlying distributions converges to zero. We have that

$$d_w(W,Z) = \sup_{h\in BL(\mathbb{R})}|\mathbb{E}h(W) - \mathbb{E}h(Z)| \tag{23}$$

#### D.1.1   INTRODUCTION TO STEIN'S METHOD FOR NORMAL APPROXIMATION

Stein Stein et al. (1986) introduced a powerful technique to estimate the rate of convergence of sums of weakly dependent r.v.s to the standard normal distribution. A remarkable feature of Stein's method is that it can be applied in many circumstances where dependence plays a role, therefore we propose an adaptation of Stein's method to our setting of dynamic GSCs.

Given a standard normal r.v. $Z$, Stein's lemma (stated below) provides a characterization of $Z$'s distribution.

**Lemma 5** (Stein's Lemma Chen et al. (2010)). *If $W$ has a standard normal distribution, then*

$$\mathbb{E}f'(W) = \mathbb{E}[Wf(W)], \tag{24}$$

*for all absolutely continuous functions $f : \mathbb{R} \to \mathbb{R}$ with $\mathbb{E}|f'(Z)| < \infty$. Conversely, if Equation 24 holds for all bounded, continuous, and piecewise continuously differentiable functions $f$ with $\mathbb{E}|f'(Z)| < \infty$, then $W$ has a standard normal distribution.*

In order to show that a r.v. $W$ has a distribution *close to* that of a target distribution of $Z$, one must compare the values of expectations of the two distributions on some collection of bounded functions $h : \mathbb{R} \to \mathbb{R}$. Here, Stein's lemma (Lemma 5) shows that $W \overset{d}{=} Z$, if

$$\mathbb{E}f'(W) - \mathbb{E}[Wf(W)] = 0 \tag{25}$$

holds. Observe that if the distribution of $W$ is close to that of $Z$'s distribution, then evaluating the L.H.S of Equation 25 when $W$ is replaced by $Z$ would result in a small value. Putting these difference equations together, the following linear differential equation known as *Stein's equation* is arrived at

$$f'(W) - Wf(W) = h(W) - \mathbb{E}h(Z) \tag{26}$$

The $f$ that satisfy Equation 26 with $h \in BL(\mathbb{R})$ must satisfy the following conditions for all $y, z \in \mathbb{R}$

$$\|f\| \leq 2 \,, \|f'\| \leq 2 \,, \|f''\| \leq \sqrt{2/\pi} \tag{27}$$
$$|f'(y+z) - f'(y)| \leq D|z|$$

where $h_0(y) = h(y) - \mathbb{E}h(Z)$, $c_1 = \sup_{x\geq 0}\xi(x)$, $c_2 = \sup_{x\geq 0}x(1-x\xi(x))$, and $\xi(x) = (1-\Phi)/\phi$ (where $\Phi(x)$ is the *distribution function* and $\phi(x) = \Phi'(x)$). Then, $D = (c_1 + c_2)\|h_0\|_\infty + 2$ is

a constant. Additionally, we have a bound on the covariance, given the dependent r.v.'s are also bounded.

$$\text{If } \Pr\{|X| \leq C_1\} = \Pr\{|Y| \leq C_2\} = 1, \text{ then} \tag{28}$$
$$|\operatorname{Cov}(X,Y)| \leq 4C_1 C_2 \alpha(r)$$

We take a similar approach to Sarkar et. al. Sarkar et al. (2014a) in terms of using the Wasserstein distance to bound the normal approximation. We first define the dependency in our GSCs and then propose a notion of $\alpha$ *mixing* in our context of GSCs. We obtain a tighter bound than the bound proposed in Sarkar et al. (2014a), by instead following an approach proposed by Sunklodas Sunklodas (2007).

### D.2 GAUSSIAN APPROXIMATION FOR DEPENDENT VARIABLES WITH GSC

In our model, we assume the r.v. $A_i$ to represent a $d$-simplex $\sigma_i^{(d)}$ in a GSC $\mathcal{G}$. In order to have a notion of $\alpha$-mixing in our setting, we must first define a distance between two $d$-simplices $\sigma_i$ and $\sigma_j$. We drop the $(d)$ superscript for brevity and ease of notation. We define this distance as the *Hausdorff* distance between simplices as

$$d_H(\sigma_i, \sigma_j) = \max \left\{ \sup_{v \in \sigma_i} \inf_{v' \in \sigma_j} d_g(v, v'), \sup_{v' \in \sigma_j} \inf_{v \in \sigma_i} d_g(v, v') \right\} \tag{29}$$

where $d_g(v, v')$ counts the number of edges in the geodesic connecting vertices $v$ and $v'$ in $\mathcal{G}_-^{(1)}$.

In Stein's method, the sum of dependent r.v.'s is studied by breaking the sum $Z_n$ into two sets based on the $r$ in mixing coefficient $\alpha(r)$. In our setting, given a fixed $d$-simplex $\sigma_i$, we study two partial sums pertaining to: 1) all $d$-simplices that are at most $r$-apart from $A_i$ and 2) the remaining partial sum after removing the variables pertaining to 1) from $Z_n$.

With this notion of distance between sets of r.v.'s, we modify with slight deviations from *proposition 4* in Sunklodas et. al. Sunklodas (2007) to accommodate our $\alpha$-mixing in Markov chains based on GSCs. For a sequence of r.v.'s $X_1, X_2, \cdots$ satisfying the $\alpha$-mixing condition, we write

$$Z_n = \sum_{i=1}^n A_i, \quad A_i = \frac{X_i}{B_n}, \quad B_n^2 = \mathbb{E}(\sum_{i=0}^n X_i)^2$$

We assume $B_n > 0$.

$T_i^{(m)}$ denotes the contribution of $d$-simplices that are further than $m$ away from $\sigma_i$ and $x(\sigma_i, r)$ denotes the partial sum of r.v.'s representing simplices that are exactly $r$ away from $A_i$. Therefore, $\sum_{r=0}^m x(\sigma_i, r)$ gets us all those $d$-simplices that are greater than or equal to $m$ away from $\sigma_i$. We are interested in the contribution of simplices $r$ away from $\sigma_i$ as we vary $r$ from 0 to $m$. With Proposition 5, we proceed to derive an upper bound on $d_w(Z_n, N)$ (i.e., the Wasserstein distance between $Z_n$ and $N$).

**Proposition 5.** *Let $S(\sigma_i, r)$ denote the set of $d$-simplices whose Hausdorff distance equals $r$. More formally,*

$$S(\sigma_i, r) = \{\sigma_j : d_H(\sigma_i, \sigma_j) = r\}$$

*Additionally, let $\widehat{X}$ denote a* mean-centered *version of r.v. $X$. Then,*

$$x(\sigma_i, r) = \sum_{p \in S(\sigma_i, r)} A_p \qquad (x(\sigma_i, 0) = A_i)$$

$$T_i^{(m)} = Z_n - \sum_{r=0}^m x(\sigma_i, r), \quad m = 0, 1, \cdots, \qquad (T_i^{(-1)} = Z_n)$$

*Suppose that $\mathbb{E}Z_n = 0$, $\mathbb{E}Z_n^2 = 1$, and $\mathbb{E}A_i^2 < \infty$ for all $i = 1, \cdots, n$. Let $\epsilon$ be a r.v. uniformly distributed in $[0, 1]$ and independent of other r.v.'s. Let $f : \mathbb{R} \to \mathbb{R}$ be a differentiable function such that $\sup_{x \in \mathbb{R}} |f'(x)| < \infty$. Then we have*

$$\mathbb{E}f'(Z_n) - \mathbb{E}Z_n f(Z_n) = E_1 + \cdots + E_7$$

*where*

$$E_1 = -\sum_{i=1}^{n} \sum_{r \geq 1} \mathbb{E} A_i x(\sigma_i, r) \left[ f'(T_i^{(r)} + \epsilon x(\sigma_i, r)) - f'(T_i^{(r)}) \right]$$

$$E_2 = -\sum_{i=1}^{n} \mathbb{E} A_i^2 \left[ f'(T_i^{(0)} + \epsilon A_i) - f'(T_i^{(0)}) \right]$$

$$E_3 = -\sum_{i=1}^{n} \sum_{r \geq 1} \sum_{q=r+1}^{2r} \mathbb{E} A_i \widehat{x(\sigma_i, r)} \delta_i^{(q)} \,, \; E_4 = -\sum_{i=1}^{n} \sum_{r \geq 1} \sum_{q \geq 2r+1} \mathbb{E} A_i \widehat{x(\sigma_i, r)} \delta_i^{(q)}$$

$$E_5 = \sum_{i=1}^{n} \sum_{r \geq 1} \mathbb{E} A_i x(\sigma_i, r) \sum_{q=0}^{r} \mathbb{E} \delta_i^{(q)}, \; E_6 = -\sum_{i=1}^{n} \sum_{q \geq 1} \mathbb{E} \widehat{A_i^2} \delta_i^{(q)}, \; E_7 = \sum_{i=0}^{n} \mathbb{E} A_i^2 \mathbb{E} \delta_i^{(q)},$$

*and*

$$\delta_i^{(q)} = f'(T_i^{(q-1)}) - f'(T_i^{(q)}).$$

**Theorem 3.** *Consider a sequence of r.v.'s $X_1, X_2, \cdots$ that satisfy $\alpha$-mixing condition (Definition 5). Let $\mathbb{E} X_i = 0$, $Pr\{|X_i| \leq L\} = 1$, for $i = 1, \cdots, n$, for some constant $L > 0$. Then, for every $h \in BL(\mathbb{R})$,*

$$d_w(Z_n, N) \leq C(S, D) \left( \sum_{i=0}^{n} \mathbb{E} |A_i|^3 + \frac{nL^3}{B_n^3} \sum_{r=1}^{n-1} r\alpha(r) \right)$$

*where $C(S, D)$ is a finite constant which depends on $D$ and $\max_i |S(\sigma_i, r)|$.*

*Proof of Theorem 3.* Given a $d$-simplex $\sigma_i$ and its corresponding r.v. $A_i$, recall that $S(\sigma_i, r)$ denotes the set of $d$-simplices that are at Hausdorff distance $r$ away from $\sigma_i$. We additionally define $S_m$ to denote the maximum cardinality of $S(\sigma_i, r)$ for all $i$ and a fixed $r$, i.e.,

$$S_m = \max_i |S(\sigma_i, r)|$$

In order to upper bound the Wasserstein distance between $Z_n$ and $N$, we estimate the difference $\mathbb{E} h(Z_n) - \mathbb{E} h(N)$ using Proposition 5. It was shown in Proposition 5 that this difference is a sum of terms $E_1, \cdots, E_7$. We will proceed by individually bounding each term.

**Bounding $E_1$:**

$$|E_1| = \left| \sum_{i=1}^{n} \sum_{r \geq 1} \mathbb{E} A_i x(\sigma_i, r) \underbrace{\left[ f'(T_i^{(r)} + \epsilon x(\sigma_i, r)) - f'(T_i^{(r)}) \right]}_{(i)} \right| \tag{30}$$

We can upper bound term $(i)$ in Equation 30 using Equation 27 by

$$D|\epsilon||x(\sigma_i, r)| \leq D|x(\sigma_i, r)| \qquad \text{(since } |\epsilon| \leq 1)$$

Then,

$$\left| x(\sigma_i, r) \left[ f'(T_i^{(r)} + \epsilon x(\sigma_i, r)) - f'(T_i^{(r)}) \right] \right| \leq D(x(\sigma_i, r))^2 \tag{31}$$

Now, we upper bound $x(\sigma_i, r)$ as

$$x(\sigma_i, r) \leq S_m \left( \frac{L}{B_n} \right) \tag{32}$$

since each normalized r.v. in $Z_n$ is upper bounded by $L/B_n$. The L.H.S. of Equation 31 is upper bounded by $D S_m^2 \left( \frac{L^2}{B_n^2} \right)$.

We know that

$$\left| \text{Cov} \left( \underbrace{A_i}_{}, \underbrace{x(\sigma_i, r) \left[ f'(T_i^{(r)} + \epsilon x(\sigma_i, r)) - f'(T_i^{(r)}) \right]}_{} \right) \right| \tag{33}$$

$$= \underbrace{\mathbb{E}\left[A_i x(\sigma_i, r)\left[f'(T_i^{(r)} + \epsilon x(\sigma_i, r)) - f'(T_i^{(r)})\right]\right]}_{(ii)}$$

$$+ \underbrace{\mathbb{E}A_i}_{=0} \mathbb{E}x(\sigma_i, r)\left[f'(T_i^{(r)} + \epsilon x(\sigma_i, r)) - f'(T_i^{(r)})\right]$$

Notice that term $(ii)$ is nothing but the summand in Equation 30. We apply the covariance bounds (Equation 28), to obtain

$$(ii) \leq 4\left(\frac{L}{B_n}\right)\left(DS_m^2 \frac{L^2}{B_n^2}\right)\alpha(r) \leq 4DS_m^2 \frac{L^3}{B_n^3}\alpha(r) \tag{34}$$

Then, we have that

$$|E_1| \leq \sum_{i=1}^{n}\sum_{r \geq 1}\left(4DS_m^2 \frac{L^3}{B_n^3}\alpha(r)\right) \leq 4DS_m^2 \frac{nL^3}{B_n^3}\sum_{r=1}^{n-1}\alpha(r). \tag{35}$$

Here, the summation $\sum_{r=1}^{n-1}$ appears, because there are only $n$ variables that should measure the dependence between each other.

**Bounding $E_2$:**

$$|E_2| = \left|\sum_{i=1}^{n}\mathbb{E}A_i^2 \underbrace{\left[f'(T_i^{(0)} + \epsilon A_i) - f'(T_i^{(0)})\right]}_{(iii)}\right| \tag{36}$$

Using Equation 27, we have that term $(iii) \leq D|A_i|$. Then, $|E_2|$ can simply be bounded as

$$|E_2| \leq D\sum_{i=1}^{n}\mathbb{E}|A_i|^3 \tag{37}$$

**Bounding $E_3$:**

$$|E_3| = \left|\sum_{i=1}^{n}\sum_{r \geq 1}\sum_{q=r+1}^{2r}\mathbb{E}A_i\widehat{x(\sigma_i, r)}\delta_i^{(q)}\right| \tag{38}$$

$$\leq \sum_{i=1}^{n}\sum_{r \geq 1}\sum_{q=r+1}^{2r}\underbrace{\left|\mathbb{E}A_i x(\sigma_i, r)\delta_i^{(q)}\right|}_{(iv)} + \sum_{i=1}^{n}\sum_{r \geq 1}\sum_{q=r+1}^{2r}\underbrace{|\mathbb{E}A_i x(\sigma_i, r)|\,\mathbb{E}\left|\delta_i^{(q)}\right|}_{(v)}$$

Let us focus on bounding terms $(iv)$ and $(v)$, separately.

*For term $(iv)$:* We can further split it as

$$\left|\mathbb{E}\underbrace{A_i}_{a}\underbrace{x(\sigma_i, r)\delta_i^{(q)}}_{b}\right| \tag{39}$$

We have previously worked out the bounds for terms $A_i$ and $x(\sigma_i, r)$. Therefore, we now focus our attention on bounding $\delta_i^{(q)}$.

$$\begin{aligned}
\delta_i^{(q)} &= f'(T_i^{(q-1)}) - f'(T_i^{(q)}) \tag{40}\\
&= f'(T_i^{(q)} + x(\sigma_i, r)) - f'(T_i^{(q)})\\
&\leq D|x(\sigma_i, r)| \qquad \text{(using Equation 27)}\\
&\leq D\left(S_m \frac{L}{B_n}\right)
\end{aligned}$$

Therefore, term $b$ (in term $(iv)$) is upper bounded by $DS_m^2 \frac{L^2}{B_n^2}$.

Applying the covariance bound (Equation 28), we have that

$$(iv) \leq 4 \left( \frac{L}{B_n} \right) \left( DS_m^2 \frac{L^2}{B_n^2} \right) \alpha(r) \leq 4DS_m^2 \frac{L^3}{B_n^3} \alpha(r) \tag{41}$$

*For term $(v)$:* We have calculated some bounds previously, so we can again split $(v)$ as

$$\left| \mathbb{E} \underbrace{A_i \ x(\sigma_i, r)}_{c} \right| \mathbb{E} \underbrace{\left| \delta_i^{(q)} \right|}_{d} \leq \underbrace{\left[ 4 \left( \frac{L}{B_n} \right) \left( S_m \frac{L}{B_n} \right) \alpha(r) \right]}_{\text{using Eqn 28 on } c} \underbrace{\left[ D \left( S_m \frac{L}{B_n} \right) \right]}_{\text{using Eqn 40 on } d} \tag{42}$$

$$\leq 4DS_m^2 \frac{L^3}{B_n^3} \alpha(r)$$

Now, combining the inequalities for terms $(iv)$ and $(v)$, we have that

$$|E_3| \leq \sum_{i=1}^{n} \sum_{r \geq 1} \sum_{q=r+1}^{2r} 8DS_m^2 \frac{L^3}{B_n^3} \alpha(r) \leq 8DS_m^2 \frac{nL^3}{B_n^3} \sum_{r=1}^{n-1} r\alpha(r) \tag{43}$$

**Bounding $E_4$:** We have that

$$|E_4| = \left| \sum_{i=1}^{n} \sum_{r \geq 1} \sum_{q \geq 2r+1} \mathbb{E} A_i \widehat{x(\sigma_i, r)} \delta_i^{(q)} \right| \tag{44}$$

$$\leq 4DS_m^2 \frac{nL^3}{B_n^3} \sum_{r \geq 1} \sum_{q \geq 2r+1} \alpha(q - r)$$

$$\leq 4DS_m^2 \frac{nL^3}{B_n^3} \sum_{r=1}^{n-1} r\alpha(r)$$

because $\sum_{r \geq 1} \sum_{q \geq 2r+1} \alpha(q - r) < \sum_{r=1}^{n-1} r\alpha(r)$.

Now, using our previous bounds, the remaining terms, i.e., $|E_5|$, $|E_6|$, and $|E_7|$ are bounded as follows:

$$|E_5| \leq 4DS_m^2 \frac{nL^3}{B_n^3} \sum_{r=1}^{n-1} r\alpha(r) \tag{45}$$

$$|E_6| \leq 4DS_m^2 \frac{nL^3}{B_n^3} \sum_{r=1}^{n-1} \alpha(r)$$

$$|E_7| \leq D \sum_{i=0}^{n} \mathbb{E}|A_i|^3$$

Finally, we combine these bounds to arrive at our final upper bound as

$$|\mathbb{E} f'(Z_n) - \mathbb{E} Z_n f(Z_n)| \leq C(S_m, D) \left( \sum_{i=0}^{n} \mathbb{E}|A_i|^3 + \frac{nL^3}{B_n^3} \sum_{r=1}^{n-1} r\alpha(r) \right) \tag{46}$$

where $C(S_m, D)$ is a constant term depending on constants $K_m$ and $D$. This completes our proof. $\square$

## D.3 DEFERRED PROOF

In this section, we establish our estimator's result on the asymptotic normality.

Let $n = |\mathcal{G}_t(d)|$ denote the total number of $d$-simplices in GSC $\mathcal{G}_t$ at time $t$. Recall from Equation 13 that

$$\widehat{h}_T(t) := \frac{1}{n} \sum_{j=1}^{n} |P_{t+1}^{\tau}(\sigma_j^{(d)}, F)|$$

$$\widehat{d}_T(t) := \frac{1}{n} \sum_{j=1}^{n} |P_{t+1}(\sigma_j^{(d)}, F)|$$

Also, from Equation 14, we have

$$q_t := [\widehat{h}_T(t) - g\widehat{d}_T(t)] - \mathbb{E}[\widehat{h}_T(t) - g\widehat{d}_T(t) \mid S_C]$$

Additionally, let us define a variable $p_t$ for convenience as follows

$$p_t := [\widehat{h}_T(t) - g\widehat{d}_T(t)] - \mathbb{E}[\widehat{h}_T(t) - g\widehat{d}_T(t) \mid \mathcal{E}(T_C), S_C]$$

Note that $p_t$ is conditioned on both $\mathcal{E}(T_C)$ and $S_C$, while $q_t$ is just conditioned on $S_C$. Keeping these expressions in mind, we begin by showing the weak convergence with $p_t$.

**Lemma 6.** *Under Assumption 1, given $p_t$ for any finite $T_C$ and $\sigma_c > 0$*

$$\sum_{t \geq T_C+M} p_t/\sqrt{T} \overset{d}{\to} \mathcal{N}(0, \sigma_c^2) \quad \text{conditioned on } \mathcal{E}(T_C) \cap S_C \text{ as } T \to \infty$$

*Proof of Lemma 6.* Given a normalized r.v. $W_T$ which is a sum of weakly dependent r.v.'s as

$$W_T := \frac{\sum_{t \geq T_C+M} p_t}{\sqrt{\text{Var}\left(\sum_{t \geq T_C+M} p_t \mid \mathcal{E}(T_C), S_C\right)}}$$

where $p_t$ is already *bounded* and *mean-centered*. We have to show that our upper bound in Theorem 3 converges to zero, so that according to Stein's lemma 5, we have

$$W_T \overset{d}{\to} \mathcal{N}(0, 1) \quad \text{conditioned on } \mathcal{E}(T_C) \cap S_C$$

Recall that by Lemma 5.7 in Sarkar et al. (2014a), we have

$$\text{Var}\left(\sum_{t \geq T_C+M} p_t \mid \mathcal{E}(T_C), S_C\right)/T \to \sigma_c^2$$

Now, we show that conditioned on event $\mathcal{E}(T_C) \cap S_C$, our bound in Theorem 3 has a convergence rate of $O(T^{-1/2})$.

Note that our $p_t$ corresponds to $A_i$ and $T$ to $n$ in Theorem 3. As $p_t$ is a function of $S_t$, it involves $p+1$ GSCs $(\mathcal{G}_{t-p+1}, \cdots, \mathcal{G}_{t+1})$ and the distance between the $i$ and $j$-th GSC is defined as $dist(i, j) = \max(|i - j| - (p + 1), 0)$. Therefore, you will observe that we have for $x(\sigma_i, r)$ only 2 states that are at distance $r$ apart from $\sigma_i$, i.e., $A_i - r$ and $A_i + r$. Therefore, $S_m = O(1)$.

Given that $\Pr\{|p_t| \leq L\} = 1$, for $t = 1, \cdots, T$ and for some constant bound $L > 0$, we have

$$\sum_{t=1}^{T} \mathbb{E}|p_t|^3 + T\frac{L^3}{B_T^3} \sum_{r=1}^{T-1} r\alpha(r) \qquad \text{(ignoring some constant terms)} \tag{47}$$

$$\leq T\frac{L^3}{B_T^3} \left(1 + \sum_{r=1}^{T-1} r\alpha(r)\right)$$

We additionally impose that $\sum_{r=1}^{\infty} r\alpha(r) < \infty$ because we use a *decaying function* for $\alpha(\cdot)$ and $B_T^2 \geq c_0 T$ with a positive constant $c_0$. We choose

$$\alpha(r) \leq C_1 \frac{1}{r^2 (\log r)^p},$$

for $r \geq 2$ and fixed $p > 1$, where $0 < C_1 < \infty$ is a constant. Then,

$$T \frac{L^3}{B_T^3} \left( 1 + \sum_{r=1}^{T-1} r\alpha(r) \right) \leq \underbrace{\frac{L^3}{c_0^{3/2}} \left( 1 + \alpha(1) + C_1 \sum_{r=2}^{\infty} \frac{1}{r(\log r)^p} \right)}_{=K(L,C_1,c_0)<\infty} \frac{1}{\sqrt{T}} \tag{48}$$

where $K(L, C_1, c_0)$ is a positive and finite constant only depending on the quantities within the parenthesis. The inequality follows the fact $B_T^2 \geq c_0 T$. Thus, we achieve an asymptotic bound of $O(T^{-1/2})$. This completes our proof. $\qquad\square$

The weak convergence with $p_t$ implies the weak convergence with $q_t$, by a simple application of Lemma 7.2 in Sarkar et al. (2014a).

**Lemma 7** (Lemma 7.2 in Sarkar et al. (2014a)). *Suppose Lemma 6 holds. Then, under Assumption 1 and assuming $\sigma_c > 0$,*

$$\text{Conditioned on } S_C, \qquad \sum_t q_t/\sqrt{T} \xrightarrow{d} \mathcal{N}(0, \sigma_c^2) \qquad \text{as } T \to \infty$$

We now prove the weak convergence of our estimator.

*Proof of Theorem 2.* By the definition of $\widehat{g}_T - g$ in Theorem 1, we achieve

$$\sqrt{T}(\widetilde{g}_T - g) = \sqrt{T}(\widetilde{g}_T - \widehat{g}_T) + \sqrt{T}(\widehat{g}_T - g)$$
$$= O(\sqrt{T}\beta) + \frac{T^{-1/2}\sum_t q_t}{\widehat{d}_T} + \frac{T^{1/2}B_T\text{E}[\widehat{d}_T|S_C]}{\widehat{d}_T}.$$

We are able to assess the convergence of each item. Since Lemma 3 shows $\text{E}[\widehat{d}_T|S_C] \to R(C)$ and Lemma 2 shows $\text{Var}(\widehat{d}_T|S_C) \to 0$, we obtain $\widehat{d}_T \xrightarrow{p} R(C)$. By Lemma 7, $T^{-1/2}\sum_t q_t$ converges to $\mathcal{N}(0, \sigma_c^2)$. Further, Lemma 4 proves $B_T = O(b)$. Combining these results with the Slutsky's lemma, we get the following results:

$$\sqrt{T}(\widetilde{g}_T - g) = O(\sqrt{T}\beta) + \widetilde{W}_T + O_P(\sqrt{T}b),$$

where $\widetilde{W}_T$ is a random variable such as

$$\widetilde{W}_T \xrightarrow{d} \mathcal{N}(0, \sigma_c^2/R(C)^2).$$

With the settings $\beta = o(T^{-1/2})$ and $b = o(T^{-1/2})$, we obtain the statement. $\qquad\square$

