# OpenReview forum: "Higher-order Structure Prediction in Evolving Graph Simplicial Complexes"
_ICLR.cc/2021/Conference — Reject_

### Official Review · AnonReviewer3 · 2020-10-28
**Reviewer #3**

**Rating:** 6
**Confidence:** 2

**Review:**

Summary:

-	The paper goes beyond the task of link prediction between vertices, but focuses on predicting formation of higher-order structures, which involves multiple vertices simultaneously. For instance, given that authors A, B and C have a paper together, task of inferring authors A, B, C having another paper with a fourth author D is an example of higher-order relationship.

-	More specifically, authors propose a kernel estimator which predicts the evolution of the graph through simplices and how they evolve with respect to some local neighborhoods. Their estimator considers how the graph, subsequently the simplices, have evolved within some window of history. The inference process is somewhat guided by a scoring mechanism that takes into account the past interactions between vertices.

-	The paper proves consistency of the estimator, meaning that the estimator converges to the true function in probability. Additionally, they prove asymptotic normality (in the sense of Central Limit Theorem) of the estimator.

-	The authors present numerical performance of their method against various other methods based on 4 dynamic graph datasets.

Strengths:

-	For the problem of predicting higher-order (incremental) structures in an evolving graph, authors propose a method which predicts formation of higher order simplices from existing ones, while capturing substructures between vertices of the simplices. They back their intuitions up by proving theoretical guarantees for the consistency and asymptotic normality of their estimator, which implies favorable statistical properties.
-	Authors explain the differences and similarities between existing inference strategies clearly, enabling the reader to understand for which task the proposed method is specialized.
-	Experimentally, authors argue that their method shows superior performance against various different methods, such as hyperedge-based higher-order prediction algorithms as well as heuristic, random walk and deep learning-based methods. For the task of predicting d+1-simplices from existing d-simplices, the method shows superior performance in most of the settings.

Weaknesses:

-	Section 4, which is a critical part of the paper as it presents the theoretical guarantees with respect to the proposed estimator, is rather complicated and lacks explanations of theorems, definitions and assumptions. In my opinion, this section requires further elaborations as to give some intuition. For instance, what the implications of consistency of the estimator are in practice and why $\alpha$-mixing property is necessary could be explained further.
-	It is important to distinguish between the settings for which the proposed method and the existing work are designed for. None of the existing methods were designed to predict higher-order simplices from existing lower-order simplices. Hyperedge-based methods predict interaction of a group of nodes, while Benson et. al. predict closed simplices from open simplices. Link prediction methods focus on single edge evolutions. These tasks are not aligned with each other and prior work is not designed for the experimental setup proposed in the paper. I believe the experiment does not reflect the whole potential of other methods and is biased towards the current method, but conveys the message that proposed algorithm has superior performance for the particular task in question.
-	Using history of the evolving graph requires additional storage load, which is not discussed in details. It is argued that the proposed method has comparable/improved run-time performance, but space complexity is also very crucial, especially when graphs have growing number of edges and many vertices. In my opinion, this is an important ground of comparison which should be presented with more details.

Additional Comments/Questions:

-	Why would you give higher score when two vertices co-occur in a higher dimensional simplex in the past?
-	 How would you compare storage requirements of your approach to existing higher-order prediction algorithms as in Benson et. al. 2018 or hypergraph-based approaches?
-	Definition of sub-complex includes $B_{t,k}$ but the value of $k$ is not specified. Do you pick a single, fixed $k$ across all definition? This is not properly defined and creates a confusion.
-	Section “4.1 Consistency” has a confusing presentation of definitions and concepts. Definition of $S_C$ is not clearly presented and I believe bias/variance terms for the estimator may need further intuitions and explanations.
-	I couldn’t see any insight for how information is encoded by the proposed method. Specifically, how are simplices (closed simplices) inferred from the dataset? How are open and closed simplices are distinguished from the raw data?

Score:

I vote for accepting the paper, but my decision is borderline. I haven’t checked the proofs in details but I appreciate the analytical effort that was put into this paper. Having consistency and asymptotic normality results strengthens the claims for the proposed estimator. However, I am not totally sure in which real-world tasks this method has significant advantages and would outperform existing methods because the experimental setting is designed for the task of inferring (d+1)-simplices from d-simplices. I am open for further discussion with the authors regarding my concerns and their future comments/responses.

---

> ### Author Response · Authors · 2020-11-18
> **Thank you for your detailed comments. We have tried to addressed your concern. (1/2)**
>
> - Weakness
>
> **R3Q1**: Section 4, which is a critical part of the paper as it presents the theoretical guarantees with respect to the proposed estimator, is rather complicated and lacks explanations of theorems, definitions and assumptions. In my opinion, this section requires further elaborations to give some intuition. For instance, what the implications of consistency of the estimator are in practice and why α-mixing property is necessary could be explained further.
>
> **R3A1**: Thank you for the suggestion. The consistency states that the errors in our estimator converge to zero with many samples. Alpha-mixing is that the phenomenon approaches independence as $T \rightarrow \infty$. We have added explanations for both, with new paragraphs and figures in the main body.
>
>
> **R3Q2**: It is important to distinguish between the settings for which the proposed method and the existing work are designed for. None of the existing methods were designed to predict higher-order simplices from existing lower-order simplices. Hyperedge-based methods predict interaction of a group of nodes, while Benson et. al. predict closed simplices from open simplices. Link prediction methods focus on single edge evolutions. These tasks are not aligned with each other and prior work is not designed for the experimental setup proposed in the paper. I believe the experiment does not reflect the whole potential of other methods and is biased towards the current method, but conveys the message that the proposed algorithm has superior performance for the particular task in question.
>
> **R3A2**: Higher-order structure prediction is a highly complex combinatorial task which is further complicated when dealing with evolving graphs. There does not exist a single definition of higher-order prediction on which there is clear consensus. The aim of our experiments is to highlight the benefits of modeling higher-order relationships as simplices in our GSC and conduct experiments on popular real world datasets to show their benefits. To this effect, we compare our method to methods that were most closely related to ours. Having said this, we agree that there is a mismatch between the problems being solved by the different methods and we have added explicit statements to the experiments section to highlight this point. Thanks!
>
>
> **R3Q3**: Using history of the evolving graph requires additional storage load, which is not discussed in detail. It is argued that the proposed method has comparable/improved run-time performance, but space complexity is also very crucial, especially when graphs have growing number of edges and many vertices. In my opinion, this is an important ground of comparison which should be presented with more details.
>
> **R3A3**: We have accordingly modified our draft by adding a new storage complexity analysis of our estimator.
>
> - Additional Comments/Questions:
>
> **R3Q4**: Why would you give higher score when two vertices co-occur in a higher dimensional simplex in the past?
>
> **R3A4**: This is a heuristic approach very similar to the one employed by the “preferential attachment” method for single link prediction, whereby vertices that are known to co-occur in past relationships are more likely to be part of future relationships too. Many similar scoring functions, such as counting common neighbors among two vertices, are quite commonly used in many heuristic based link prediction algorithms and work quite well in practise.
>
>
> **R3Q5**: How would you compare storage requirements of your approach to existing higher-order prediction algorithms as in Benson et. al. 2018 or hypergraph-based approaches?
>
> **R3A5**: We are not entirely sure of the exact storage requirements of these competing approaches, but the dominating cost in their methods would be the storage of hyper-edges in hypergraph methods and storage of open/closed triangles in case of Benson et. al. Both these would heavily depend on the number of hyperedges and open/closed triangles per dataset and there is no worst case upper bound that is proposed by either of these methods on the number of hyperedges and triangles.
> We would like to point out that for a hypergraph $H$ to model what a simplicial complex does, for any given hyperedge $\sigma \in H$, all its possible subsets $\tau \subseteq \sigma$, must also be present in $H$. Therefore, it is easy to see that a single simplex with say $n$ vertices in our GSC model, would require $2^n$ “separate and explicit” hyperedges in hypergraph methods, which is exponential in $n$.
>
>
> **R3Q6**: Definition of sub-complex includes Bt,k but the value of k is not specified. Do you pick a single, fixed $k$ across all definition? This is not properly defined and creates a confusion.
>
> **R3A6**: Yes, $k$ is fixed.

---

> > ### Author Response · Authors · 2020-11-18
> > **Thank you for your detailed comments. We have tried to addressed your concern. (2/2)**
> >
> >
> >
> > **R3Q7**: Section “4.1 Consistency” has a confusing presentation of definitions and concepts. Definition of $S_C$ is not clearly presented and I believe bias/variance terms for the estimator may need further intuitions and explanations.
> >
> > **R3A7**: About the bias-variance decomposition, we decompose the prediction error into bias (model expressive power) and variance (uncertainty of algorithm), and analyze them by parts. By this individual analysis, we can investigate the whole prediction error. It is a very common approach in statistical learning theory. We also added an additional description in the main body to make this clearer to the readers.
> >
> >
> > **R3Q8**: I couldn’t see any insight for how information is encoded by the proposed method. Specifically, how are simplices (closed simplices) inferred from the dataset? How are open and closed simplices are distinguished from the raw data?
> >
> > **R3A8**: The dataset includes timestamped simplices per line of the file. E.g., “100 {1,2,3}” denotes the arrival of a triangle / 2-simplex {1,2,3} at timestamp 100. This is also mentioned in the “Datasets” section in our experiments.
> >
> >
> > - Score:
> >
> > **R3Q9**: I vote for accepting the paper, but my decision is borderline. I haven’t checked the proofs in details but I appreciate the analytical effort that was put into this paper. Having consistency and asymptotic normality results strengthens the claims for the proposed estimator. However, I am not totally sure in which real-world tasks this method has significant advantages and would outperform existing methods because the experimental setting is designed for the task of inferring $(d+1)$-simplices from $d$-simplices. I am open for further discussion with the authors regarding my concerns and their future comments/responses.
> >
> > **R3A9**: We would like to refer you to **R4A1**’s examples to see real-world situations in which our GSC prediction model would have significant advantages over existing methods. This is also added to the Introduction of our revised draft.

---

> > > ### Comment · AnonReviewer3 · 2020-11-23
> > > **Thank you for clarifying my comments and explaining the subtleties**
> > >
> > > **R3A1, R3A2, R3A6, R3A7, R3A8:** Thank you for the clear answers and the respective modifications made in the main text.
> > >
> > > **R3A3:**  Your storage complexity ignores the storage of history of the evolution of the graph, which may have significant impact on the storage complexity, to the best of my understanding.
> > >
> > > **R3A4:** Thank you for this clarification, this explanation makes your scoring strategy clearer for me.
> > >
> > > **R3A5** If some simplex structure is dissolved over the course of time in an evolving graph, this requires your GSC to encode individual links, as well. However, I think the argument about hypergraphs and how they encode simplices is a good point for distinguishing your method against hypergraphs with respect to storage advantages.
> > >
> > > **R3A9** I have read the new examples that were attached to the introduction of the paper. I could understand the motivation to be provided by these examples to clarify the importance of detecting high-order relationships among graphs. However, I don’t think thalidomide example has clear connections to your GSC. Your method infers formation of higher-order simplices from existing ones. However, the specific example of thalidomide is not related to simplicial structures formed among atoms; it is rather related to detecting symmetry of the compound. I don’t see how this example is related to your GSC. If I am missing some subtle point, I will be happy if you could point it out for me.

---

> > > > ### Author Response · Authors · 2020-11-24
> > > > **Our responses**
> > > >
> > > > Thanks for your clear feedback. Below, we answer your queries.
> > > >
> > > > **R3A3**: Your storage complexity ignores the storage of history of the evolution of the graph, which may have significant impact on the storage complexity, to the best of my understanding.
> > > >
> > > > **Ans**: Yes, if we took into account the storage cost to hold the evolving graphs per time-window of size p, prior to our feature generation itself, then we would incur an additional storage cost of $O(|\mathcal{G}_{t,p}|)$. We left this term out because we were focusing on just the storage cost of our estimator post feature generation. But, we can certainly add this term in our storage complexity analysis, if needed.
> > > >
> > > >
> > > > **R3A9** I have read the new examples that were attached to the introduction of the paper. I could understand the motivation to be provided by these examples to clarify the importance of detecting high-order relationships among graphs. However, I don’t think thalidomide example has clear connections to your GSC. Your method infers formation of higher-order simplices from existing ones. However, the specific example of thalidomide is not related to simplicial structures formed among atoms; it is rather related to detecting symmetry of the compound. I don’t see how this example is related to your GSC. If I am missing some subtle point, I will be happy if you could point it out for me.
> > > >
> > > > **Ans**: That is a very good point. On further thought, we find ourselves in agreement with your observation about the thalidomide example being more related to higher-order structures and chiral symmetry, and not so much about predicting new higher-order groups.
> > > >
> > > > Nevertheless, there is a slight difference between the R and S configurations that is worth noting. One of the bonds attached to the N atom, is spatially located on the **top of the plane** while the other is located at the **bottom of the plane**, which makes this particular bond the one that differentiates the two configurations.
> > > > More here: https://en.wikipedia.org/wiki/Absolute_configuration
> > > >
> > > > We understand that this real-world example is **not directly connected** to our GSCs in their present state and problem setting, but we do believe that our work could be a significant first step in a direction where someone could, for example, make use of **“weighted GSCs”** or **“vertex/edge-colored GSCs”** to capture the bond’s spatial properties as well, in order to make such subtle differentiations, that are necessary for real-world problems to work. Of course, one could argue that this can also be done with weighted hypergraphs, but we do have better compression properties.
> > > >
> > > > We found an alternative example from stereochemistry related to **Vitamin B-12**, which is the largest and most structurally complex vitamin, where our GSC prediction problem finds more relevance.
> > > >
> > > > If the reviewers arrive at some consensus, we are more than happy to remove the thalidomide example completely or replace it with an example based on Vitamin B12 in our revised draft.

---

### Official Review · AnonReviewer4 · 2020-10-29
**Presents an estimator for higher-order structure prediction in graphs and provide asymptotic guarantees for it.**

**Rating:** 6
**Confidence:** 2

**Review:**

This paper presents an estimator that predict higher-order structure in time-varying graphs. The authors present an kernel-based estimator, prove that it is consistent when the indicator variable for whether a particular (d+1)-dimensional simplex is Bernoulli distributed with a function g. The authors prove that their estimator is asymptotically normal.  The authors also present some experiments on real-world data

Some comments and questions:

1. A concrete example that motivates this problem would be useful in the introduction. As someone who hadn't encountered this problem previously I found it hard to keep a motivational example in my mind while reading this paper.

2. In equation 7, should it be |\tilde{g}_T -g|, that is, is the absolute value missing? Otherwise, I dont see why \tilde{g}_T - g represents the error here. Relatedly it would be nice if the authors provide some intuition behind why the two terms in the equation correspond to the variance and bias respectively.

3. It would be nice if the authors could give an example of a scenario where the process is alpha-mixing.

4. Proposition 1 makes a reference to Theorem 3 (which is deep within the appendix), mentioning this in the proposition statement would be nice.

5. The statement of Theorem 2 conditions of S_C which is undefined.

---

> ### Author Response · Authors · 2020-11-18
> **Thank you for your warm comment. We have tried to address your concerns.**
>
> **R4Q1**: A concrete example that motivates this problem would be useful in the introduction. As someone who hadn't encountered this problem previously I found it hard to keep a motivational example in my mind while reading this paper.
>
> **R4A1**: We refer you to the Examples outlined in **R2A1**. But for the sake of readability, we reiterate the main points here.
>
> **Following examples (also added to Introduction in updated draft)**
>
> In **Organic and organo-metallic synthesis (Organic Chemistry)**, it is quite common to have the SAME set of elements interacting with each other in different configurations, termed as “confirmations” in Chemistry, which result in very different functioning compounds. For e.g., R-thalidomide and S-thalidomide are two different confirmations of thalidomide, where the R-form was meant to help sedate pregnant women, while the S-form unfortunately resulted in birth defects. This is a famous example in stereo chemistry, referred to as the “Thalidomide Tragedy”.
> (More → https://sites.science.oregonstate.edu/~gablek/CH334/Chapter5/Thalidomide.htm )
> This example is widely cited in structural chemistry to show the consequences of mistaking two extremely close configurations (differing by a single bond) as being the same. Structure prediction in drug synthesis allows chemists to achieve a much higher yield and avoid wastage of expensive resources.
>
> **Gene expression networks** have nodes that represent genes and edges connect genes which have similar expression patterns. Subgraphs called "modules" are tightly connected genes in such a gene expression network. Genomics research provides evidence that higher-order gene expression relationships (like second and third-order) and their measurements can have very important implications for cancer prognosis.
>
> Papers:
> - Shuangge Ma, Michael R Kosorok, Jian Huang, and Ying Dai,Incorporating higher-order representative features improves prediction in network-based cancer prognosis analysis in BMC Med Genomics, 2011
> - Zhang B, Horvath S. A general framework for weighted gene co-expression network analysis. Statistical Applications in Genetics and Molecular Biology. 2005;
>
> When making structural predictions in these aforementioned examples, our simplicial complex based approach provides much more fine-grained control over competing methods by capturing subtler differences in configurations.
>
> **R4Q2**: In equation 7, should it be $|\tilde{g}_T -g|$, that is, is the absolute value missing? Otherwise, I don't see why $\tilde{g}_T - g$ represents the error here.
>
> **R4A2**: Absolute values are not required, because we approximate the difference by the normal distribution in terms of asymptotic normality. The analogy is that when discussing the central limit theorem (CLT), we analyze the difference between the expected value and the sample mean, without absolute values.
>
> **R4Q3**: Relatedly it would be nice if the authors provide some intuition behind why the two terms in the equation correspond to the variance and bias respectively.
>
> **R4A3**: Thank you for your suggestion. Intuitively, we decompose the error into bias (model expressive power) and variance (uncertainty of algorithm), and analyze them by parts. By this individual analysis, we can investigate the whole prediction error. It is a very common technique used in statistical learning theory. We have also added additional descriptions in the main body.
>
> **R4Q4**: It would be nice if the authors could give an example of a scenario where the process is alpha-mixing.
>
> **R4A4**:  The simplest example is the evolution of stock prices in financial markets. The movement of stock prices today does not correlate with the movement of stock prices 10 years ago. Such a structure is represented by alpha-mixing. We added a brief explanation of alpha-mixing and also provided an example in the updated draft.
>
> **R4Q5**: Proposition 1 makes a reference to Theorem 3 (which is deep within the appendix), mentioning this in the proposition statement would be nice.
>
> **R4A5**: We added an explanation about the statement.
>
> **R4Q6**: The statement of Theorem 2 conditions of $S_C$ which is undefined.
>
> **R4A6**: Thank you for this point. It was actually defined at the beginning of the Theory section, however from your comment it is possible that readers might miss this definition. Therefore, we decided to repeat the definition prior to Theorem 2 as well.

---

### Official Review · AnonReviewer2 · 2020-10-31
**The paper develops a nonparametric kernel estimator based method for high-order structure prediction in evolving graphs.**

**Rating:** 4
**Confidence:** 4

**Review:**

This paper provide a method for high-order structure prediction problem. Specifically, the paper first defines a high-order structure on graphs named graph simplicial complex (GSC). Then the paper introduces a feature generation method used for the high-order structures. The features are also used in the proposed method for high-order structure prediction. The proposed method is based on a nonparametric kernel which carries the feature similarities of high-order structures. With this kernel the method uses a Bernoulli distribution for the prediction of the existence of the high-order structure in unseen times.

In my opinion, the paper has the following strengths and weaknesses.

For strengths, first, the kernel estimator based method does not require learning process and shows good efficiency in the prediction tasks. The kernel estimator also plays an important role in capturing the high-order interactions in the evolving graphs. Second, the paper provides with theoretical analysis on the consistency and asymptotic normality of the proposed kernel estimator. The theories show possibility of inferring the estimation error and the confidence intervals for predictions.

For weaknesses, first, the paper ought to make further clarification on the essential differences (and maybe even better, the advantages) between the defined GSC with the traditional high-order structures, such as simplex, hyperedges, or just small graphs. Also experimentally the paper should show more of the advantages of using GSCs. For example, the experiment only use d=1 and 2. Perhaps the experiment should show its advantage when dealing with much higher-order structure predictions rather than these simple cases. Second, the presentation of the paper needs further polish. The paper defines notations on-the-fly, and it makes the readers not easy to follow. For example, if we only look at the notations related to G, there are G_t, G_{-}^{(d)}, G_{t,p}, G'_t(), G_t(d), G_{t-}^{(d)}, and they are defined here and there in Section 3. The readers will need to make their own notation table just to follow the paper. I personally failed at looking for the definition of G_{t-}^{(d)}. In Definition 3, it looks like it is a vertex set, but in the time complexity analysis it becomes a number.  Also, in the time complexity analysis, does the complexity of counting number of simplices of each d need to be considered? Third, for the experiments, the paper should consider using more dynamic methods of link prediction especially when you are only comparing the results of prediction lower-order structures. In Table 1, what is the runtime for baselines, training or making predictions? Overall the experiments need to be more concrete. Last but not least, this paper might not be a good fit for ICLR community since it does not focus on the representation learning methods.

---

> ### Author Response · Authors · 2020-11-18
> **Thank you for your kind comment. We have done our best to address your points. (1/2)**
>
> **R2Q1**: For weaknesses, first, the paper ought to make further clarification on the essential differences (and maybe even better, the advantages) between the defined GSC with the traditional high-order structures, such as simplex, hyperedges, or just small graphs.
>
> **R2A1**: The advantage of the GSC over other methods are as follows.
> 1. The higher dimensional analogue of link prediction in graphs is naturally the structure prediction on GSCs, because vertices and edges of the graph are nothing but 0- and 1-simplices in our GSC, which then allows for higher dimensional simplices to be introduced as well.
> 2. As highlighted in our introduction, the actual relationship between vertices of a single hyperedge (especially with large cardinality) are unknown, whereas this is easily fixed with a simplicial complex in our GSC, thus providing us much more fine-grained control over the many possible higher-order relationships.
> 3. The outputs/artifacts of a higher-order relationship can be captured using a hypergraph, but the finer details of “who interacted with whom” is best captured by our GSC model.
>
> **Following examples (also added to Introduction in updated draft)**
>
> In **Organic and organo-metallic synthesis (Organic Chemistry)**, it is quite common to have the SAME set of elements interacting with each other in different configurations, termed as “confirmations” in Chemistry, which result in very different functioning compounds. For e.g., R-thalidomide and S-thalidomide are two different confirmations of thalidomide, where the R-form was meant to help sedate pregnant women, while the S-form unfortunately resulted in birth defects. This is a famous example in stereo chemistry, referred to as the “Thalidomide Tragedy”.
> (More → https://sites.science.oregonstate.edu/~gablek/CH334/Chapter5/Thalidomide.htm )
> This example is widely cited in structural chemistry to show the consequences of mistaking two extremely close configurations (differing by a single bond) as being the same. Structure prediction in drug synthesis allows chemists to achieve a much higher yield and avoid wastage of expensive resources.
>
> **Gene expression networks** have nodes that represent genes and edges connect genes which have similar expression patterns. Subgraphs called "modules" are tightly connected genes in such a gene expression network. Genomics research provides evidence that higher-order gene expression relationships (like second and third-order) and their measurements can have very important implications for cancer prognosis.
>
> Papers:
> - Shuangge Ma, Michael R Kosorok, Jian Huang, and Ying Dai,Incorporating higher-order representative features improves prediction in network-based cancer prognosis analysis in BMC Med Genomics, 2011
> - Zhang B, Horvath S. A general framework for weighted gene co-expression network analysis. Statistical Applications in Genetics and Molecular Biology. 2005;
>
> When making structural predictions in these aforementioned examples, our simplicial complex based approach provides much more fine-grained control over competing methods by capturing subtler differences in configurations.
>
>
>
> **R2Q2**: Also experimentally the paper should show more of the advantages of using GSCs. For example, the experiment only used d=1 and 2. Perhaps the experiment should show its advantage when dealing with much higher-order structure predictions rather than these simple cases.
>
> **R2A2**: Thanks for pointing this out. We conducted experiments with higher dimensional simplex predictions (upto d=8) and the results can be found in our updated draft in Figure 3. We observe that the gains in AUC achieved exhibit submodularity (diminishing returns property), i.e., the marginal gains in AUC scores while predicting the arrival of higher dimensional simplices diminishes as the dimensions increase.
>
> **R2Q3**: Second, the presentation of the paper needs further polish. The paper defines notations on-the-fly, and it makes the readers not easy to follow. For example, if we only look at the notations related to $G$, there are $G_t, G_{-}^{(d)}, G_{t,p}, G't(), G_t(d), G_{t-}^{(d)}$, and they are defined here and there in Section 3. The readers will need to make their own notation table just to follow the paper.
>
> **R2A3**: Thank you for the nice suggestion. We followed your advice and made a notation table in the main body. We hope the table will improve the readability of our method.
>
> **R2Q4**: I personally failed at looking for the definition of $G_{t-}^{(d)}$. In Definition 3, it looks like it is a vertex set, but in the time complexity analysis it becomes a number.
>
> **R2A4**: Thanks for pointing this mistake out, it was a typo. This was meant to be $|G_{t-}^{(d)}|$, which we fixed now.

---

> > ### Author Response · Authors · 2020-11-18
> > **Thank you for your kind comment. We have done our best to address your points. (2/2)**
> >
> >
> > **R2Q5**: Also, in the time complexity analysis, does the complexity of counting the number of simplices of each d need to be considered?
> >
> > **R2A5**: No, they all needn’t be considered. Just the $d-1$ and $d$-simplices suffice in the estimator calculations.
> >
> > **R2Q6**: Third, for the experiments, the paper should consider using more dynamic methods of link prediction especially when you are only comparing the results of prediction lower-order structures. In Table 1, what is the runtime for baselines, training or making predictions? Overall the experiments need to be more concrete.
> >
> > **R2A6**: Unless we have mistaken your query / comment, all our prediction experiments are carried out on dynamically evolving graphs (maybe you can further clarify?). Table 1 shows the runtimes that include the time to train and also predict.
> >
> >
> > **R2Q7**: Last but not least, this paper might not be a good fit for the ICLR community since it does not focus on the representation learning methods.
> >
> > **R2A5**: We respectfully disagree with this opinion. Our method follows an approach grounded in statistical learning theory to learn representations for higher-order relationships (via simplices) and similar relationships in their neighborhood (captured by face-vector feature representations) in a graph that evolves over time. Additionally, our method comes with theoretical guarantees like consistency and convergence that make it attractive and very apt for a venue like ICLR.

---

> > > ### Comment · AnonReviewer2 · 2020-11-18
> > > **Thanks for the responses. Further comments are as follows.**
> > >
> > > R2A1: Thanks for the response. Can you further clarify the differences (or even better, the advantages) between the proposed GSC over subgraphs of unseen times? Can the GSCs be simply seen as the hyperedges with links inside (or subgraphs)?
> > >
> > > R2A2 & R2A6: Thanks for the additional experimental results. What I original mean was the comparison between the proposed method and the baselines in higher-order cases. Also, in lower-order cases (such as d = 1), dynamic methods/algorithms for link prediction can be used as baselines. Based on the experimental settings of the paper, the dynamic methods/algorithms could also be adapted for comparison with the proposed method when d > 1. Please take these experiments into consideration.
> > >
> > > R2A3 & A4: Thanks for adding the notation table. Can you further clarify if G_{t-}^{(d)} and G_{t, -}^{(d)} are the same? Also, in the newly added Table 1, why is G_{t, -}^{(0)} used in the definition of B_{t,k}(i)? Isn't the superscript (d) on G denoting the dimension of the simplices?
> > >
> > > R2A7: IMHO, the original comments still hold for the same reason. ICLR has more preference on the deep learning/representation learning-related work. Based on the proposed techniques and the quality of the paper, the Web Conference or SIGKDD might be a better fit.

---

> > > > ### Author Response · Authors · 2020-11-22
> > > > **Reply to Further Comments**
> > > >
> > > > **R2A1**: Thanks for the response. Can you further clarify the differences (or even better, the advantages) between the proposed GSC over subgraphs of unseen times? Can the GSCs be simply seen as the hyperedges with links inside (or subgraphs)?
> > > >
> > > > **Ans**: We hope the following will help clear your confusion.
> > > >
> > > > i) **comparing against subgraphs/graphs:** we reiterate that subgraphs, or for that matter graphs, CANNOT adequately and unambiguously model higher-order relationships. Given a graph $G=(V,E)$, where V={A,B,C} and E={ {A,B}, {B,C}, {A,C} }, such a graph (or subgraph) cannot clearly indicate whether this is a relationship with ONLY pairwise relationships (i.e., {A,B}, {B,C} and {A,C}), also called “open triangle” in Benson et. al. OR if this is {A,B,C} all simultaneously participating in a relationship (closed triangle). This problem doesn’t arise when using a simplicial complex, because the former situation is modelled by three 1-simplices (edge) and the latter by a single 2-simplex (triangle).
> > > >
> > > > ii) **comparing against hypergraphs:** (we refer you to part of **R3A5**) We would like to point out that for a hypergraph $H$ to model what a simplicial complex does, for any given hyperedge $\sigma$ in $H$, all its possible subsets $\tau \subseteq \sigma$, must also be present in $H$. Therefore, it is easy to see that a single simplex with say $n$ vertices in our GSC model, would require $2^n$ “separate and explicit” hyperedges in hypergraph methods, which is exponential in $n$. Therefore, a hypergraph cannot achieve what our GSC does in a much compressed manner, because a simplex includes all its lower-dimensional simplex (faces).
> > > >
> > > > **R2A2 & R2A6**: Thanks for the additional experimental results. What I original mean was the comparison between the proposed method and the baselines in higher-order cases. Also, in lower-order cases (such as d = 1), dynamic methods/algorithms for link prediction can be used as baselines. Based on the experimental settings of the paper, the dynamic methods/algorithms could also be adapted for comparison with the proposed method when d > 1. Please take these experiments into consideration.
> > > >
> > > > **Ans**: Thanks. We have modified two existing deep learning methods that deal with single link prediction in temporal dynamic networks.
> > > > Namely,
> > > >
> > > > **TT**: “Inductive Representation Learning on Temporal Graphs” by Xu et.al. (ICLR 2020)
> > > >
> > > > **TN**: “Temporal Graph Networks for Deep learning on Dynamic graphs” by Rossi et. al. (ICML 2020 Workshop on Graph Representation Learning)
> > > >
> > > > Both these methods use deep learning models and work for predicting the arrival of single future links in dynamic networks. They had been evaluated on Reddit, Twitter and Wikipedia. Note that we have evaluated them on our dynamic datasets + averaged their single link predictions to predict higher order links. Results of our comparison are shown in Table 2 of our revised draft. Not surprisingly, we find that our method outperforms these methods as well in our given setting.
> > > >
> > > > **R2A3 & A4**: Thanks for adding the notation table. Can you further clarify if G_{t-}^{(d)} and G_{t, -}^{(d)} are the same? Also, in the newly added Table 1, why is G_{t, -}^{(0)} used in the definition of B_{t,k}(i)? Isn't the superscript (d) on G denoting the dimension of the simplices?
> > > >
> > > > **Ans**: Thank you for the detailed reading.
> > > >
> > > > (i) $G_{t-}^{(d)}$ and $G_{t, -}^{(d)}$ are the same. We unified the notation.
> > > >
> > > > (ii) $B_{t,k}(i)$ depends on $G_{t-}^{(0)}$, because $G_{t-}^{(0)}$ is the set of $0$-dimensional simplices (i.e., vertex set of the graph) (already written in the paragraph before Definition 3). So, this is natural.
> > > >
> > > > **R2A7**: IMHO, the original comments still hold for the same reason. ICLR has more preference on the deep learning/representation learning-related work. Based on the proposed techniques and the quality of the paper, the Web Conference or SIGKDD might be a better fit.
> > > >
> > > > **Ans**: This is a matter of personal opinion. Anyway, we decided to submit our paper to ICLR for the following reasons: (i) there are many good papers in ICLR that don't use deep learning, and (ii) ICLR community is much more familiar with statistical learning theory than SIGKDD or WebConf.
> > > >
> > > > **Some examples of non-deep learning papers in ICLR:**
> > > >
> > > > https://openreview.net/pdf?id=r1lGO0EKDH
> > > >
> > > > https://openreview.net/pdf?id=HyxJ1xBYDH
> > > >
> > > > https://openreview.net/pdf?id=BkG5SjR5YQ
> > > >
> > > > https://openreview.net/pdf?id=H1lhqpEYPr
> > > >
> > > > **Examples of statistical learning theory papers in ICLR:**
> > > >
> > > > https://openreview.net/pdf?id=ByeGzlrKwH
> > > >
> > > > https://openreview.net/pdf?id=r1e_FpNFDr
> > > >
> > > > https://openreview.net/pdf?id=rkgg6xBYDH

---

### Author Response · Authors · 2020-11-24
**General comment to all reviewers**

We would like to thank all the anonymous reviewers for their insightful comments, which have contributed to the improvement of our draft, via the addition of new examples, experimental comparisons, notation tables, and intuitive explanations around our theoretical material, to name a few modifications.

---

### Decision · Program_Chairs · 2021-01-11
**Final Decision**

**Decision:**

Reject

**Comment:**

This paper proposes a method for predicting higher-order structure in time-varying graphs. The paper was reviewed by three expert reviewers, and while they expressed appreciation for the sensible solution, they have remaining concerns about the novel contributions and comparisons (analytical and empirical) with previous approaches. Also, the paper would be clearer if examples are used to illustrate the important points of the paper. The authors are encouraged to continue research, taking into consideration the detailed comments provided by the reviewers.